# AQUILA: Communication Efficient Federated Learning with Adaptive Quantization of Lazily-Aggregated Gradients

## Abstract

The development and deployment of federated learning (FL) have been bottlenecked by the heavy communication overheads of high-dimensional models between the distributed client nodes and the central server. To achieve better error-communication tradeoffs, recent efforts have been made to either adaptively reduce the communication frequency by skipping unimportant updates, a.k.a. lazily-aggregated quantization (LAQ), or adjust the quantization bits for each communication. In this paper, we propose a unifying communication efficient framework for FL based on **a**daptive **qu**ant**i**zation of **l**azily-**a**ggregated gradients (**AQUILA**), which adaptively adjusts two mutually-dependent factors, the communication frequency and the quantization level, in a synergistic way. Specifically, we start from a careful investigation on the classical LAQ scheme and formulate AQUILA as an optimization problem where the optimal quantization level per communication is selected by minimizing the gradient loss caused by updates skipping. Meanwhile, we adjust the LAQ strategy to better fit the novel quantization criterion and thus keep the communication frequency at an appropriate level. The effectiveness and convergence of the proposed AQUILA framework are theoretically verified. The experimental results demonstrate that AQUILA can reduce around **50%** of overall transmitted bits compared to existing methods while achieving the same level of model accuracy in a number of non-homogeneous FL scenarios, including Non-IID data distribution and heterogeneous model architecture. The proposed AQUILA is highly adaptive and compatible to existing FL settings.

## 1 Introduction

With the deployment of ubiquitous sensing and computing devices, the Internet of things (IoT) as well as many other distributed systems have gradually grown from concept to reality, bringing dramatic convenience to people's daily life (Du et al., 2020; Liu et al., 2020a; Hard et al., 2018). To fully utilize such distributed computing resources, distributed learning provides a promising framework that can achieve comparable performance with the traditional centralized learning scheme. However, the privacy and security of sensitive data during the updating and transmitting processes in distributed learning have been a growing concern. In this context, *federated learning (FL)* (McMahan et al., 2017; Yang et al., 2019) has been developed, enabling distributed devices to collaboratively learn a global model without privacy leakage by keeping private data sets isolated and masking transmitted information with secure approaches like differential privacy (Abadi et al., 2016), secret sharing techniques (Bonawitz et al., 2017) and homomorphic encryption (Liu et al., 2020b). Due to its privacy and security preserving property and great potentials in some distributed but privacy-sensitive fields like finance and health, FL has attracted tremendous attentions from both academia and industry in recent years.

Unfortunately, in many FL applications like image classification and objective recognition, the models to be trained tend to be high-dimensional, which lead to heavy communication overheads, for example, a Resnet-152 network has over 58 million parameters (He et al., 2016). Hence, communication efficiency has become one of the key bottlenecks of FL. To this end, recent researches have tried to reduce the communication frequency, for example, Sun et al. (2020) proposes the lazily-aggregated quantization (LAQ) method to reduce communication rounds by skipping some

unnecessary parameter uploads. To further reduce transmitted bits per communication, LAQ can be used jointly with gradient compression techniques, e.g. quantization and sparsification (Strom, 2015; Wangni et al., 2018; Lin et al., 2018; Han et al., 2020). Moreover, Mao et al. (2021) develops the Adaptive Quantized Gradient (AQG) for LAQ to adjust the quantization bit among multiple given levels during training. However, the AQG is not sufficiently adaptive, for example, in the two-level AQG with 4 bit and 2 bit, the situation of 3 bit and 1 bit is not covered at all. In a separate line of work, Jhunjhunwala et al. (2021) develops an adaptive quantization rule (AdaQuantFL) for FL which can search in a given range for an optimal quantization level and achieve a better error-communication tradeoff.

Previous work has investigated how to optimize communication frequency or adjust quantization levels in a highly adaptive fashion, but not both. Intuitively, we ask the question, *can we adaptively adjust the quantization level in LAQ to further reduce communication rounds and transmitted bits simultaneously?* A straightforward approach is to train LAQ jointly with state-of-the-art adaptive quantization methods like AdaQuantFL. However, quantization level and communication frequency are mutually dependent in establishing the model convergence and must be adjusted together. For example, lazy aggregation leads to the skip of some gradients that no longer require quantization, while the choice of quantization levels directly affects the quality of a gradient update and thus whether it is selected for transmission and aggregation. Therefore, the key question of our work is how to jointly leverage these two complementary yet mutually-dependent degrees of freedom for further optimizing communication efficiency in FL.

The key idea of this paper is to select the optimal quantization bits for each communication round in LAQ by optimizing the gradient loss caused by skipping quantized updates, which gives a novel quantization criterion that can cooperate with LAQ strategy to further reduce overall transmitted bits while maintaining the desired convergence properties of LAQ. The contributions of this paper are trifold and summarized as follows. 1) We propose a FL framework with **a**daptive **qu**ant**i**zation of **l**azily-**a**ggregated gradients termed AQUILA, which simultaneously adjusts the communication frequency and quantization level in a synergistic fashion. 2) We formulate AQUILA as an optimization problem and develop an upperbound for the gradient loss caused by communication skipping, which gives a novel adaptive quantization criterion which is theoretically proven to be more efficient compared to AdaQuantFL while maintaining the same convergence properties. 3) We experimentally evaluate the performance of AQUILA in a number of non-homogeneous FL settings, including Non-IID data distribution and various heterogeneous model architecture. The experimental results show that AQUILA can significantly mitigate the communication overhead over a number of baselines including fixed-bit LAQ and the naive combination of LAQ and AdaQuantFL. Our approach is highly adaptive and compatible to existing FL settings.

## 2 BACKGROUND AND RELATED WORK

Consider a FL system with one central server and a clients set $\mathbb{M}$ of $M$ distributed clients to collaboratively train a global model parameterized by $\boldsymbol{\theta}^*$. At iteration $k$, each client $m \in \mathbb{M}$ first trains the global model $\boldsymbol{\theta}^k$ on its local data $D_m^k$, and sends the local gradient $\boldsymbol{g}_m^k = \nabla f_m(D_m^k; \boldsymbol{\theta}^k)$ to the central server. Then the server aggregates the parameters and updates the global parameter by:

$$\boldsymbol{\theta}^{k+1} = \boldsymbol{\theta}^k - \frac{\alpha}{M} \sum_{m \in \mathbb{M}} \boldsymbol{g}_m^k. \tag{1}$$

To reduce communication overheads with gradient quantization, the stochastic uniform quantizer (Alistarh et al., 2017) is usually adopted. For any local gradient $\boldsymbol{g} \in \mathbb{R}^d$, the quantized value of its $i$-th dimension with quantization level $b$ is defined as:

$$Q_b(g_i) = \|\boldsymbol{g}\|_2 \cdot \text{sign}(g_i) \cdot \xi_i(\boldsymbol{g}, b), \tag{2}$$

where $\xi_i(\boldsymbol{g}, b)$ is a random variable defined as follows. Let $l \in \{0, 1, 2, ..., b-1\}$ be an integer satisfying $|g_i| / \|\boldsymbol{g}\|_2 \in [l/b, (l+1)/b)$, then:

$$\xi_i(\boldsymbol{g}, b) = \begin{cases} (\ell+1)/b & \text{with probability } (b \cdot \|g_i\|)/\|\boldsymbol{g}\|_2 - l \\ \ell/b & \text{otherwise.} \end{cases} \tag{3}$$

It is clear that with quantization level $b$, the number of bits for transmitting a quantized gradient from a client to the central server is $C_b = d\lceil \log_2(b+1)\rceil + d + 32$, with 32 bits for $\|g\|_2$, 1 bit for each sign $(g_i)$, and $\log_2(b+1)$ bits for each $\xi_i(\boldsymbol{g}, b)$.

For communication rounds reduction, the LAQ proposes to let the client $m \in \mathbb{M}$ upload its newly-quantized local gradient $Q_b(\boldsymbol{g}_m^k)$ at iteration $k$ only when the change in local gradient is sufficiently large, i.e.,

$$\left\|Q_b(\boldsymbol{g}_m^k) - Q_b(\hat{\boldsymbol{g}}_m^{k-1})\right\|_2^2 \geq \frac{\sum_{d=1}^{D} \xi_d \left\|\boldsymbol{\theta}^{k+1-d} - \boldsymbol{\theta}^{k-d}\right\|_2^2}{\alpha^2 M^2} + 3(\|\varepsilon_b(\hat{\boldsymbol{g}}_m^{k-1})\|_2^2 + \left\|\varepsilon_b(\boldsymbol{g}_m^k)\right\|_2^2), \quad (4)$$

where $Q_b(\hat{\boldsymbol{g}}_m^{k-1})$ is the last quantized upload from client $m$, $\varepsilon_b(\hat{\boldsymbol{g}}_m^{k-1})$ and $\varepsilon_b(\boldsymbol{g}_m^k)$ denote quantization errors, and $\{\xi_d\}_{d=1}^{D}$ are some predetermined constant weights. Notice here a **fixed** quantization level $b$ is used. In LAQ, if the difference between client $m$'s newly-quantized local gradient $Q_b(\boldsymbol{g}_m^k)$ and the last upload is smaller than a threshold involving quantization errors and global model's innovation, client $m$ will skip the upload of $Q_b(\boldsymbol{g}_m^k)$ at iteration $k$ and the central server will reuses $Q_b(\hat{\boldsymbol{g}}_m^{k-1})$ for such lazy aggregation:

$$\hat{\boldsymbol{\theta}}^{k+1} = \boldsymbol{\theta}^k - \frac{\alpha}{M}\sum_{m\in\mathbb{M}} Q_b(\hat{\boldsymbol{g}}_m^k) = \boldsymbol{\theta}^k - \frac{\alpha}{M}\sum_{m\in\mathbb{M}\backslash\mathbb{M}_0^k} Q_b(\boldsymbol{g}_m^k) - \frac{\alpha}{M}\sum_{m\in\mathbb{M}_0^k} Q_b(\hat{\boldsymbol{g}}_m^{k-1}), \quad (5)$$

where $\mathbb{M}_0^k$ denotes the subset of clients that skip the new gradient update and reuse the old quantized gradient at iteration $k$. Besides, $\hat{\boldsymbol{g}}_m^k$ represents the actual gradient for aggregation from client $m$, which is $\boldsymbol{g}_m^k$ for $m \in \mathbb{M} \setminus \mathbb{M}_0^k$, while $\hat{\boldsymbol{g}}_m^{k-1}$ for $m \in \mathbb{M}_0^k$.

Recently, AdaQuantFL is proposed to achieve a better error-communication tradeoff by adaptively adjusting the quantization levels during FL training (Jhunjhunwala et al., 2021). Specifically, AdaQuantFL computes iteration $k$'s optimal quantization level $b_k^*$ based on the following criterion involving training loss and initial quantization level $b_0$:

$$b_k^* = \sqrt{f\left(\boldsymbol{\theta}^0\right)/f\left(\boldsymbol{\theta}^k\right)} \cdot b_0, \quad (6)$$

where $f(\boldsymbol{\theta}^0)$ and $f(\boldsymbol{\theta}^k)$ are the global training loss at iteration 0 and $k$, respectively.

However, AdaQuantFL transmits quantized gradients at **every iteration**. In order to skip unnecessary communication rounds and adaptively adjust quantization level for each communication jointly, an naive approach is to quantize lazily aggregated gradients with AdaQuantFL. Nevertheless, it fails to achieve efficient communication due to a number of reasons. Firstly, given the descending trend of training loss, AdaQuantFL's criterion (6) may lead to high quantization bit number even exceeding 32 late in the training process, which is too much for cases where the global convergence is already approaching. Secondly, higher quantization level results in smaller quantization error, which will lead to lower communication threshold in LAQ's criterion (4) and thus higher frequency of transmission. Therefore it is desirable to develop more efficient adaptive quantization method in the lazily-aggregated setting to systematically improve communication efficiency in FL.

## 3 METHOD

### 3.1 ADAPTIVE QUANTIZATION OF LAZILY-AGGREGATED GRADIENTS (AQUILA)

Given the above limitations of the naive joint use of existing adaptive quantization criterion and lazy aggregation strategy, this paper aims to design a unifying framework for communication efficiency optimization where the quantization level and communication frequency are adjusted in a synergistic and interactive way. Based on a careful investigation on LAQ, we design a novel quantization criterion where the optimal quantization level is selected by minimizing the expected gradient loss caused by skipping quantized updates. The rationale behind such strategy is, by formulating the adaptive quantization problem of lazily-aggregated gradients as optimizing the expected gradient loss with respect to the number of quantization bits, we can get an adaptive quantization criterion based on local gradient updates while maintaining or even improving the convergence property of

LAQ. To get the optimization target, we first derive an upperbound for the expected gradient loss in terms of quantization bits as elaborated in Section 3.2, which gives a novel adaptive quantization criterion (7) that selects the quantization level for client $m$ at iteration $k$ based on initial quantization level $b_0$ and the change between client $m$'s newly computed gradient $g_m^k$ and the last uploaded gradient $\hat{g}_m^{k-1}$:

$$(b_m^k)^* = b_0 \cdot \sqrt{\left\|\boldsymbol{g}_m^1 - \hat{\boldsymbol{g}}_m^0\right\|_2^2 / \left\|\boldsymbol{g}_m^k - \hat{\boldsymbol{g}}_m^{k-1}\right\|_2^2}. \tag{7}$$

The superiority of (7) comes from the following two aspects. Firstly, the gradient updates tend to fluctuate along with the training process instead of keeping descending like the loss value, and thus prevent the quantization level from increasing tremendously compared with the initial level. Secondly, with lazy aggregation criterion based on gradient updates like (4), the transmitted bits in AQUILA is further controlled, since the gradient update for actual transmission in (7) is lower bounded by the lazy aggregation criterion, and therefore high-bit transmission for small update is more likely to be skipped.

To better fit the larger quantization errors induced by fewer quantization bits in (7), we modify the communication criterion as follows to avoid the potential expansion of clients group to be skipped:

$$\left\|Q_{b_m^k}(\boldsymbol{g}_m^k) - Q_{\hat{b}_m^{k-1}}(\hat{\boldsymbol{g}}_m^{k-1})\right\|_2^2 \geq \frac{\sum_{d=1}^D \xi_d \left\|\boldsymbol{\theta}^{k+1-d} - \boldsymbol{\theta}^{k-d}\right\|_2^2}{\alpha^2 M^2} + 2\left\|\varepsilon_{b_m^k}(\boldsymbol{g}_m^k) - \varepsilon_{\hat{b}_m^{k-1}}(\hat{\boldsymbol{g}}_m^{k-1})\right\|_2^2, \tag{8}$$

where all the notations are the same as in (4) except the heterogeneous quantization level $b_m^k$ and $\hat{b}_m^{k-1}$ for each client. For detailed development of (8), please refer to the Appendix.

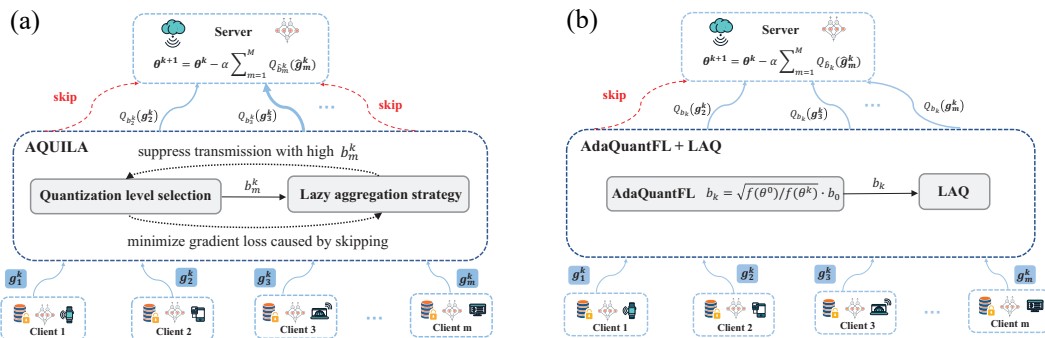

Figure 1: The schematic illustration of the communication efficient FL with AQUILA in comparison with the naive combination of AdaQuantFL and LAQ. The blue lines indicating the transmission of quantized gradients in AQUILA are drawn in different thicknesses to represent different quantization levels selected by various clients.

The cooperation of the novel adaptive quantization criterion (7) and the modified lazy aggregation strategy (8) is illustrated in Fig. 1a. Compared to the naive combination of AdaQuantFL and LAQ where the mutual influence between adaptive quantization and lazy aggregation has not been considered as shown in Fig. 1b, our AQUILA framework adaptively optimizes the allocation of quantization bits throughout training to promote the convergence of lazy aggregation, and at the same time utilizes the lazy aggregation strategy to improve the efficiency of adaptive quantization by skipping high-bit transmission. The proposed AQUILA's effect of suppressing the transmission of high quantization bits has been verified in our experiments, as shown in Fig. 20 in the Appendix. Besides, with the adjusted lazy aggregation strategy (8), AQUILA well addresses the problem of high communication frequency in the late training process of naive combination of AdaQuantFL and LAQ, as indicated by Fig. 2.

The proposed AQUILA is summarized as follows in Algorithm 1. At iteration $k = 0$, all clients are forced to transmit local gradients quantized with the initial level $b_0$. At iteration $k \in \{1, 2, ..., K\}$, the server first broadcasts the global model $\boldsymbol{\theta}^k$ to all clients. Each client $m$ computes $\boldsymbol{g}_m^k$ with local training data, and then uses it to select an optimal quantization level by (7). Then, each client

computes its gradient update after quantization and determines whether to upload the update or not based on the communication criterion (8). Finally, the server updates the new global model $\boldsymbol{\theta}^{k+1}$ with up-to-date quantized gradients $Q_{b_m^k}(g_m^k)$ for those clients who send the uploads at iteration $k$, while reusing the old quantized gradients $Q_{\hat{b}_m^{k-1}}(\hat{g}_m^{k-1})$ for those who skip the uploads.

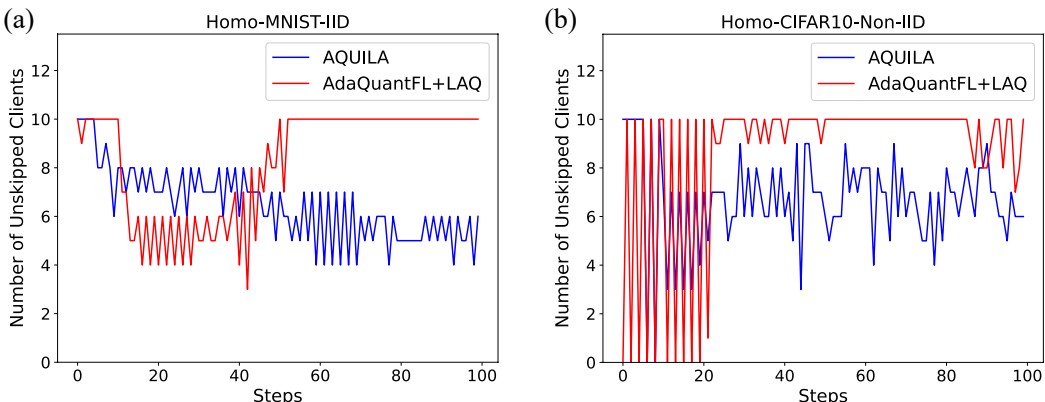

Figure 2: Comparison of AQUILA and AdaQuantFL+LAQ on the number of unskipped clients per step in three experiment settings with homogeneous model architecture. AdaQuantFL clashes with the threshold condition in LAQ and results in high communication frequency due to the increasing quantization level late in the training process, but AQUILA keeps the communication frequency at an appropriate level throughout the training.

---

**Algorithm 1** Communication Efficient FL with AQUILA

---

**Input:** the number of communication rounds $K$, the learning rate $\alpha$, the maximum communication level $b_{max}$

**Initialize:** the initial global model parameter $\boldsymbol{\theta}^0$ and the initial quantization level $b_0$.

1: Server broadcasts $\boldsymbol{\theta}^0$ to all clients.
2: **for** each client $m \in \mathbb{M}$ **in parallel do**
3:     Calculates local gradient $\boldsymbol{g}_m^0$ and sends the quantized gradient $Q_{b_0}(\boldsymbol{g}_m^0)$.
4:     Set $\hat{\boldsymbol{g}}_m^0 = \boldsymbol{g}_m^0$ and $\hat{b}_m^0 = b_0$ on both sides.
5: **end for**
6: **for** $k = 1, 2, ..., K$ **do**
7:     Server broadcasts $\boldsymbol{\theta}^k$ to all clients.
8:     **for** each client $m \in \mathbb{M}$ **in parallel do**
9:         Calculates local gradient $\boldsymbol{g}_m^k$.
10:        Computes the optimal local quantization level $b_m^k$ by (7).
11:        **if** $b_m^k \geq b_{max}$ **then**
12:           $b_m^k = b_{max}$.
13:        **end if**
14:        **if** (8) holds for client $m$ **then**
15:           Client $m$ computes and sends the quantized gradient $Q_{b_m^k}(\boldsymbol{g}_m^k)$.
16:           Set $\hat{\boldsymbol{g}}_m^k = \boldsymbol{g}_m^k$ and $\hat{b}_m^k = b_m^k$ on both sides.
17:        **else**
18:           Client $m$ sends nothing.
19:           Set $\hat{\boldsymbol{g}}_m^k = \hat{\boldsymbol{g}}_m^{k-1}$ and $\hat{b}_m^k = \hat{b}_m^{k-1}$ on both sides.
20:        **end if**
21:     **end for**
22:     Server updates $\boldsymbol{\theta}^{k+1}$ by $\boldsymbol{\theta}^k - \alpha \sum_{m=1}^{M} Q_{\hat{b}_m^k}(\hat{\boldsymbol{g}}_m^k)$.
23: **end for**

---

## 3.2 THEORETICAL DERIVATION AND ANALYSIS OF AQUILA

As mentioned before, in this work we bound the expected gradient loss caused by skipping updates with respect to quantization bits. Specifically, if the communication criterion (8) holds for client $m$ at iteration $k$, it does not contribute to iteration $k$'s gradient loss, otherwise, the loss caused by client $m$ will be minimized with the optimal quantization criterion (7). In this section, the theoretical derivation of the target upper bound is based on following standard assumptions:

**Assumption 1.** *Loss function $f(\boldsymbol{\theta}) = \sum_{m \in \mathbb{M}} f_m(\boldsymbol{\theta})$ is $L$-smooth, and $f_m(\boldsymbol{\theta})$ is $L_m$-smooth.*

**Assumption 2.** *The quantization operation is unbiased with $\mathbb{E}[Q_b(w)|w] = w$, and its variance satisfy: $\forall w \in \mathbb{R}^d, \mathbb{E}[\|Q_b(w) - w\|_2^2 |w] \le q_b \|w\|_2^2$, where $q_b$ is a positive constant.*

Here we adopt the definition of $B$ in AdaQuantFL, but modified it as follows:
**Definition 1** (Number of Bits sent from client $m$ to the server, $B_m$). *The total number of bits that has been sent from client $m$ to the central server until a given time instant is denoted by $B_m$.*
With the definition of $B_m$, the upper bound of the loss induced by gradient skipping is indicated by the following theorem:
**Theorem 1.** *Under Assumption 1 and 2, the expected gradient loss caused by lazy aggregation has the following upper bound:*

$$
\begin{aligned}
\mathbb{E}[\|\hat{\boldsymbol{\theta}}^{k+1} - \boldsymbol{\theta}^{k+1}\|_2^2] &\le \frac{3\alpha^2}{M} \sum_{m \in \mathbb{M}} \frac{d_m d \lceil \log_2(4b_m^k) \rceil}{B_m} \|\boldsymbol{g}_m^k - \hat{\boldsymbol{g}}_m^{k-1}\|_2^2 \\
&+ \frac{3\alpha^2}{M} \sum_{m \in \mathbb{M}} \left( \frac{d\sigma^2}{(\hat{b}_m^{k-1})^2} + \frac{d\sigma^2}{(b_m^k)^2} \right) + \frac{3\alpha^2}{M} \sum_{m \in \mathbb{M}} \frac{d_m(d+32)}{B_m} \|\boldsymbol{g}_m^k - \hat{\boldsymbol{g}}_m^{k-1}\|_2^2,
\end{aligned}
\tag{9}
$$

*where $\sigma^2$ denotes the maximum $\|w\|_2^2$, and $d_m$ is a positive constant determined by $L_m$. See the Appendix for proof details.*

For each client $m$, the optimal $b_m^k$ is selected by setting the first derivative of the upper bound (9) to zero:

$$
(b_m^k)^* = \sqrt{\left( 2\sigma^2 B_m \log_e(2) \right) / \left( d_m \|\boldsymbol{g}_m^k - \hat{\boldsymbol{g}}_m^{k-1}\|_2^2 \right)}.
\tag{10}
$$

Therefore, an adaptive quantization criterion which does not rely on unknown parameters such as $\sigma$ and $d_m$ can be achieved through dividing $b_m^k$ by $b_m^1$, which gives the adaptive selection criterion in (7) if $b_m^1$ for each client $m$ is chosen as the initial quantization level $b_0$.

**Comparison with AdaQuantFL.** In AdaQuantFL, the optimal quantization level is selected by minimizing the upper bound of the expected squared norm of the gradient $\nabla f(\boldsymbol{\theta}^k)$ of a non-convex objective function, and thus the convergence of AdaQuantFL is guaranteed. However, the proposed AQUILA is proven to be more efficient with sufficiently small learning rates as shown in Theorem 2. Besides, it is theoretically guaranteed that the squared norm of the global gradient $\nabla f(\boldsymbol{\theta}^k)$ is also able to converge under AQUILA's adaptive quantization criterion developed for lazy aggregation.

**Theorem 2.** *Under Assumption 1 and 2, if the learning rate $\alpha$ satisfies $\alpha \le (1/ML) \cdot \sqrt{2P\xi/L}$, the following inequality holds:*

$$
(b_m^k)^* = b_0 \sqrt{\frac{\|\boldsymbol{g}_m^1 - \hat{\boldsymbol{g}}_m^0\|_2^2}{\|\boldsymbol{g}_m^k - \hat{\boldsymbol{g}}_m^{k-1}\|_2^2}} \le b_0 \sqrt{\frac{\alpha^2 M^2 L^3 \|\boldsymbol{\theta}^1 - \boldsymbol{\theta}^0\|_2^2}{2\xi[f(\boldsymbol{\theta}^k) - f(\boldsymbol{\theta}^*)]}} \le b_0 \sqrt{\frac{f(\boldsymbol{\theta}^0) - f(\boldsymbol{\theta}^*)}{f(\boldsymbol{\theta}^k) - f(\boldsymbol{\theta}^*)}},
\tag{11}
$$

*where $\xi$ and $P$ are both positive constants. See the Appendix for proof details.*

Theorem 2 indicates that with appropriate learning rate, the proposed AQUILA uses less number of bits per communication compared to AdaQuantFL.

**Convergence analysis.** The Lyapunov function of AQUILA is defined in the same way as LAQ and AQG (Mao et al., 2021):

$$\mathbb{V}(\boldsymbol{\theta}^k) = f(\boldsymbol{\theta}^k) - f(\boldsymbol{\theta}^*) + \sum_{d=1}^{D} \sum_{j=d}^{D} \frac{\xi_j}{\alpha} \left\| \boldsymbol{\theta}^{k+1-d} - \boldsymbol{\theta}^{k-d} \right\|_2^2, \quad (12)$$

where $\boldsymbol{\theta}^*$ is the optimal solution of $\min_{\boldsymbol{\theta}} f(\boldsymbol{\theta})$.

Convergence guarantee of federated learning with lazy aggregation has been well discussed in Sun et al. (2020). More specifically, both the objective residual $f(\boldsymbol{\theta}^k) - f(\boldsymbol{\theta}^*)$ and the parameter differences term in Lyapunov function are guaranteed to descend along with the training process. Therefore, the squared norm of the gradient $\nabla f(\boldsymbol{\theta}^k)$ is also guaranteed to converge based on the $L$-smoothness assumption which results in: $\|\nabla f(\boldsymbol{\theta}^k)\|_2^2 \leq 2L[f(\boldsymbol{\theta}^k) - f(\boldsymbol{\theta}^*)]$.

# 4 EXPERIMENTS AND DISCUSSION

## 4.1 EXPERIMENT SETUP

**Dataset.** In this paper, we evaluate our method with MNIST and CIFAR10 dataset, considering both IID and Non-IID data distribution. To simulate Non-IID situation, each client is assigned with two classes of data at most and the amount of data for each class is balanced.

**Parameters.** We set total client number $M = 10$, and follow the settings in LAQ, where $D = 10$ and $\xi_1 = \xi_2 = \cdots = \xi_D = 0.8/D$. In terms of initial quantization level, a low level $b_0 = 2$ is chosen for simpler tasks with MNIST, and a larger level $b_0 = 6$ is selected for more complex tasks with CIFAR10. For both AdaQuantFL and AQUILA, we set an upperbound for quantization level as $b_{max} = 16$ in case that the level grows too high.

**Training.** We train a CNN with MNIST and a Resnet18 network with CIFAR10. The hyperparameters of our experiments are shown in Table 1 in Appendix.

We first evaluate our proposed AQUILA with homogeneous settings where all the local models share the same architecture as the global model. The performance of AQUILA is compared with several state-of-the-art methods including FedAvg (McMahan et al., 2017), QSGD (Alistarh et al., 2017), AdaQuantFL, fixed-bit LAQ and the naive combination of AdaQuantFL with LAQ. For the choice of quantization level for fixed-bit LAQ, since AQUILA's initial level $b_0$ is 2 for MNIST and 6 for CIFAR10, we compare AQUILA with LAQ-2 and LAQ-6 for MNIST and CIFAR 10 respectively. Besides, the performance of LAQ with the upperbound $b_{max} = 16$ is also evaluated.

We also evaluate our proposed AQUILA with HeteroFL (Diao et al., 2020), where the local models trained at clients' side are heterogeneous. Assume the global model at iteration $k$ is $\theta_g^k$ and its size is $d_g \cdot h_g$, then the local model of each client $m$ can be selected by $\theta_m^k = \theta_g^k [: d_m, : h_m]$, where $d_m = r_m d_g$ and $h_m = r_m h_g$ respectively. In this paper, we choose three various model complexity levels $r = \{a, b, c\} = \{1, 0.5, 0.25\}$.

Fig. 3 shows the *training loss vs total transmitted bits* curve of experiments with IID MNIST / CIFAR10 for homogeneous model architecture, IID MNIST / CIFAR10 for 100%-50% heterogeneous model architecture, and Non-IID MNIST / CIFAR10 for 100%-25% heterogeneous model architecture. The corresponding *transmitted bits vs steps* curves of the above experiment settings are shown in Fig. 4, which represents how many bits are transmitted in each step. Methods without adaptive quantization and lazy aggregation like FedAvg and QSGD are not included in Fig. 4 for simplicity. All the other experiment results are provided in Appendix, with Fig. 5 to Fig. 9 for homogeneous models, and Fig. 10 to Fig. 19 for heterogeneous models.

## 4.2 PERFORMANCE ANALYSIS

In this part, we analyze the performance of AQUILA with various experiment settings including Non-IID data distribution and heterogeneous model architecture. From figures in this paper, we can observe that:

- AQUILA achieves a significant transmission reduction on both MNIST and CIFAR10 as compared to AdaQuantFL and the naive combination of LAQ and AdaQuantFL. For instance, AQUILA saves 65.98% of transmitted bits compared with AdaQuantFL and 62.05% compared with the naive combination of LAQ and AdaQuantFL in the IID scenario with CIFAR10 dataset as shown in Fig. 3d, and other figures all show an obvious reduction in terms of the total transmitted bits required for convergence. Fig. 4 verifies that the quantization level selected by AQUILA will not continuously increase during training like AdaQuantFL and LAQ with AdaQuantFL.

- Similarly, AQUILA outperforms all fixed-level LAQ in terms of overall transmitted bits for both MNIST and CIFAR10, as shown in Fig. 3. For example, Fig. 3a indicates that AQUILA reduces 42.6% of total transmitted bits compared with LAQ-2 and 92.32% compared with LAQ-16 in MNIST. The transmission reduction is 38.76% for LAQ-6 and 78.09% for LAQ-16 in CIFAR10 as shown in Fig. 3e. Besides, Fig. 3 and Fig. 4 indicate that although LAQ with fixed but low quantization level like LAQ-2 and LAQ-4 sometimes transmit smaller amount of bits per step compared with AQUILA, they suffer from lower accuracy and slower convergence. It further verifies the necessity and effectiveness of our well-designed adaptive quantization criterion which achieves fast convergence with similar low-bit transmission but without degradation of the model performance.

- AQUILA also works well with heterogeneous local models. With the two various ways of distributing the heterogeneous local models in this paper, our proposed AQUILA still outperforms other methods by significantly reducing overall transmitted bits while maintaining the same convergence property and model accuracy. Please refer to Fig. 10 to Fig. 19 in the Appendix for more detailed information. These experimental results in non-homogeneous FL settings prove that our proposed AQUILA can be used in a more general and complicated federated learning scenarios.

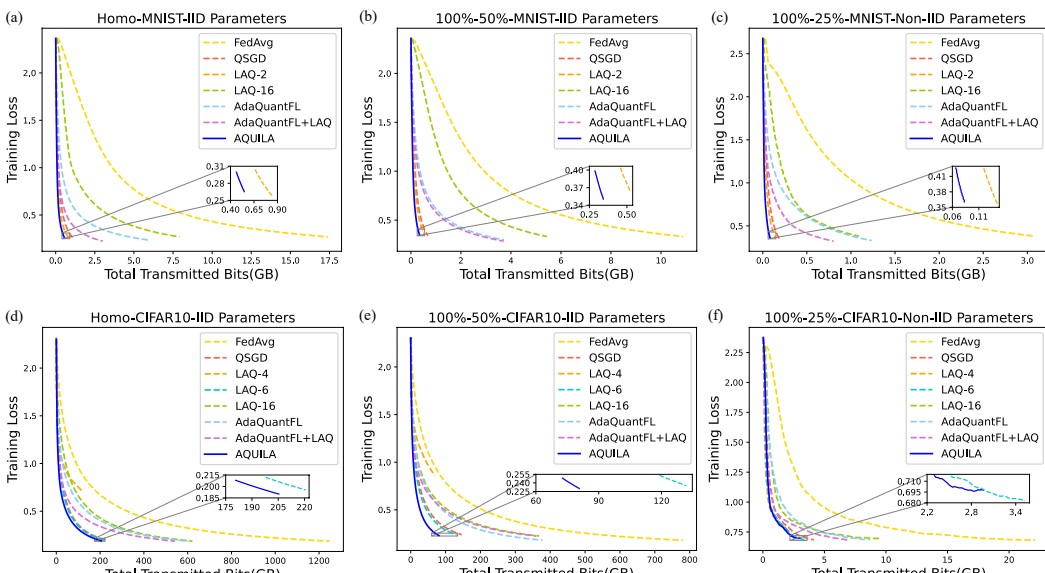

Figure 3: **Training Loss vs Total Transmitted Bits.** In this figure, LAQ-2 and LAQ-16 represent LAQ with fixed quantization level of 2 and 16 respectively. AdaQuantFL+LAQ represents the naive combination of AdaQuantFL and LAQ. For heterogeneous model architecture, 100%-50% implies that half of clients have 100% of global model, whereas the other half of clients just share $50\% * 50\%$ of the global model. Similarly, 100%-25% means the other half of clients just share $25\% * 25\%$ of the global model. Particularly, we zoom in the end of the curves to better compare AQUILA with other methods.

## 5 CONCLUSIONS AND FUTURE WORK

This paper proposes a communication efficient FL framework to simultaneously adjust two mutually-dependent degrees of freedom: communication frequency and quantization level. With the close cooperation of the novel adaptive quantization and adjusted lazy aggregation strategy developed in this paper, the proposed AQUILA has been proven to be capable of reducing the transmitted bits while maintaining the same convergence property and model performance compared against existing methods both theoretically and experimentally. The evaluation with Non-IID data distribution and various heterogeneous model architectures demonstrates that AQUILA is compatible to existing FL settings. Future works include further improvements and theoretical guarantee for FL systems with heterogeneity.

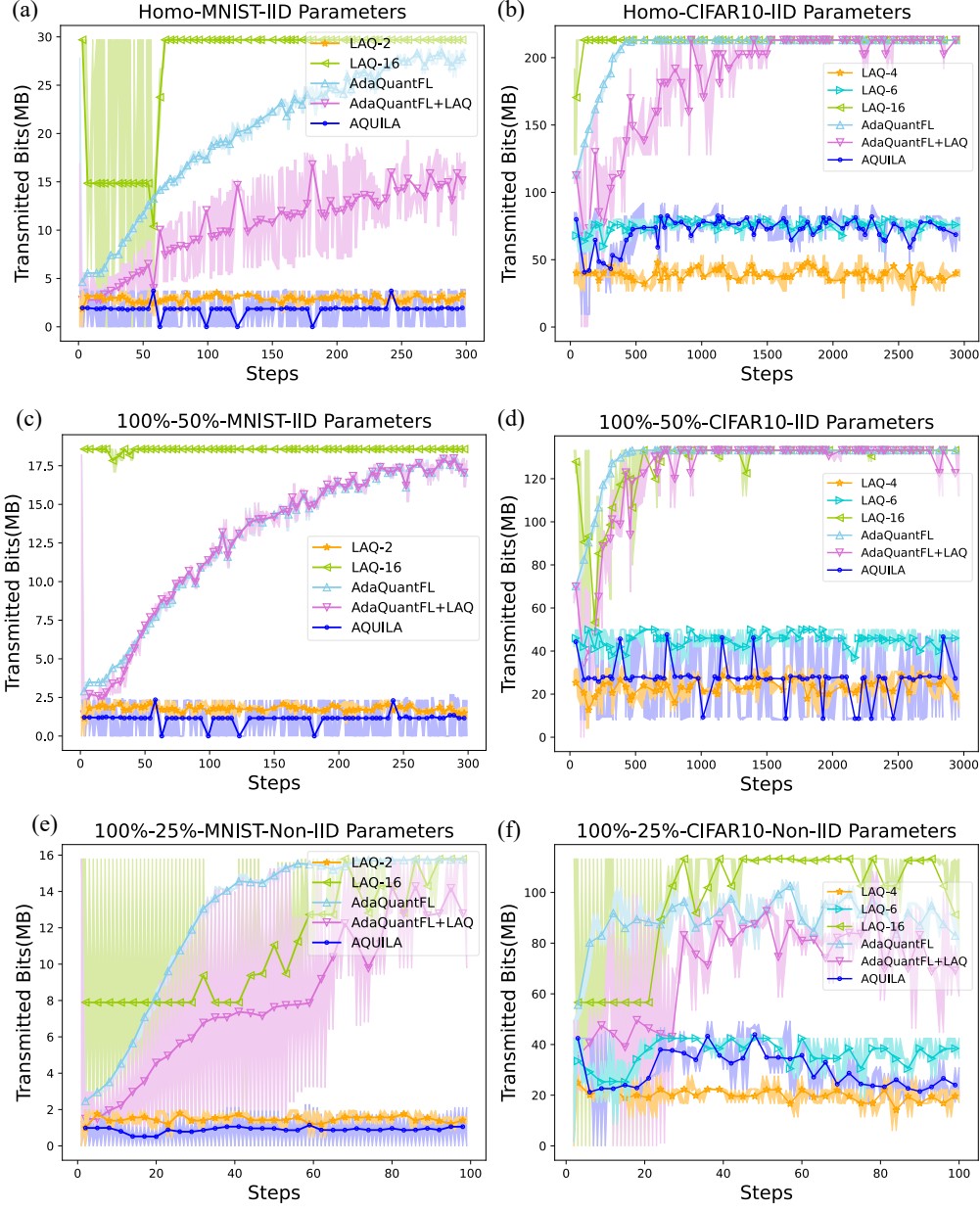

Figure 4: **Transmitted Bits vs Steps.** For better illustration, the results have been down-sampled. The solid lines represent values after down-sampling and the shadows around them represent the true values.

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

# A  APPENDIX

## A.1  SUPPLEMENTARY EXPERIMENTAL RESULTS

The appendix includes supplementary experimental results, mathematical proof of Theorem 1 and Theorem 2, and detailed derivation of the novel adaptive quantization criterion and lazy aggregation strategy. Compared to Fig. 3 and Fig. 4 in the main text, result figures in appendix show a more comprehensive evaluation with AQUILA, which contain more detailed information including but not limited to *accuracy vs steps* and *training loss vs steps* curves.

Table 1: Hyperparameters in training process

| Dataset | MNIST | | CIFAR10 | |
|---|---|---|---|---|
| Model | CNN | | ResNet18 | |
| Hidden Size | [64, 128, 256, 512] | | [64, 128, 256, 512] | |
| Data Distribution | IID | Non-IID | IID | Non-IID |
| Global Epoch $E$ | 300 | 100 | 3000 | 100 |
| Local Batch Size $B$ | 200 | 10 | 100 | 10 |
| Optimizer | SGD | SGD | SGD | SGD |
| Momentum | 0.9 | 0.9 | 0.9 | 0.9 |
| Weight Decay | 5.00E-04 | 5.00E-04 | 5.00E-04 | 5.00E-04 |
| Learning Rate $\eta$ | 0.01 | 0.01 | 0.1 | 0.1 |

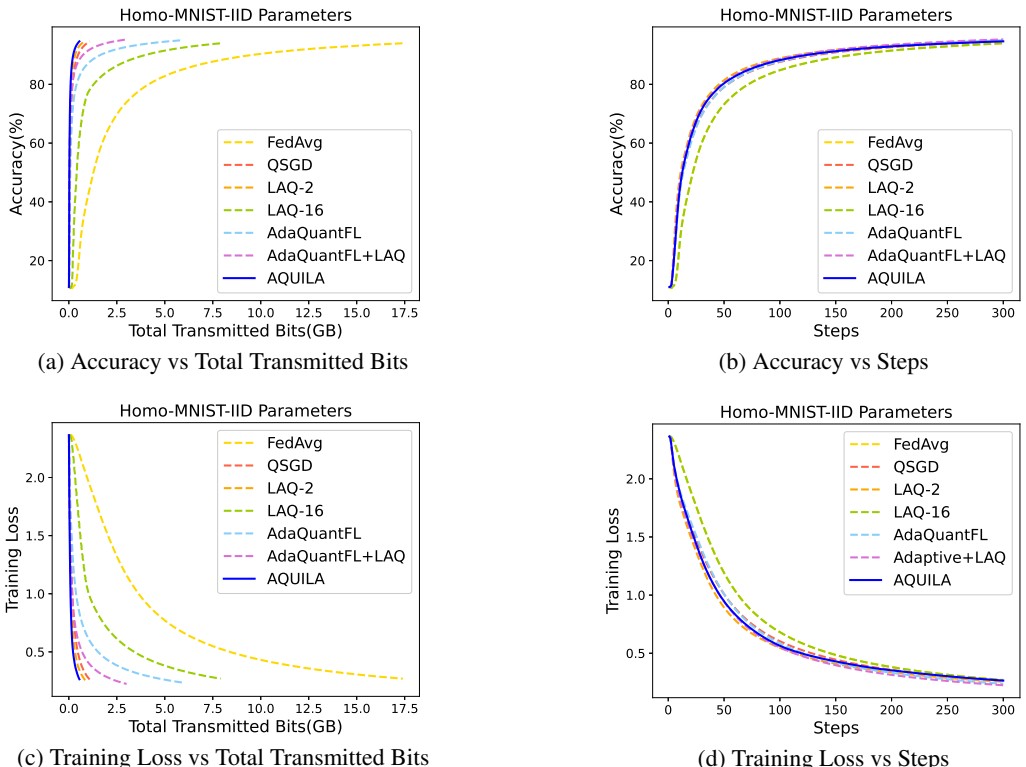

(a) Accuracy vs Total Transmitted Bits

(b) Accuracy vs Steps

(c) Training Loss vs Total Transmitted Bits

(d) Training Loss vs Steps

Figure 5: Homo-MNIST-IID

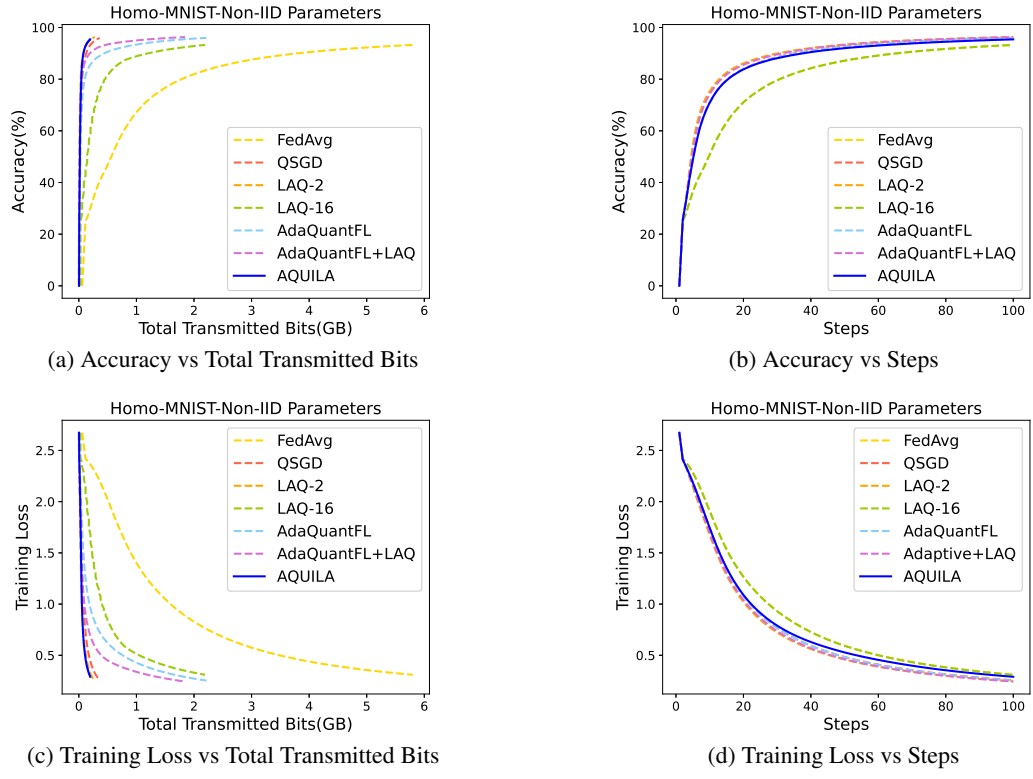

(a) Accuracy vs Total Transmitted Bits

(b) Accuracy vs Steps

(c) Training Loss vs Total Transmitted Bits

(d) Training Loss vs Steps

Figure 6: Homo-MNIST-Non-IID

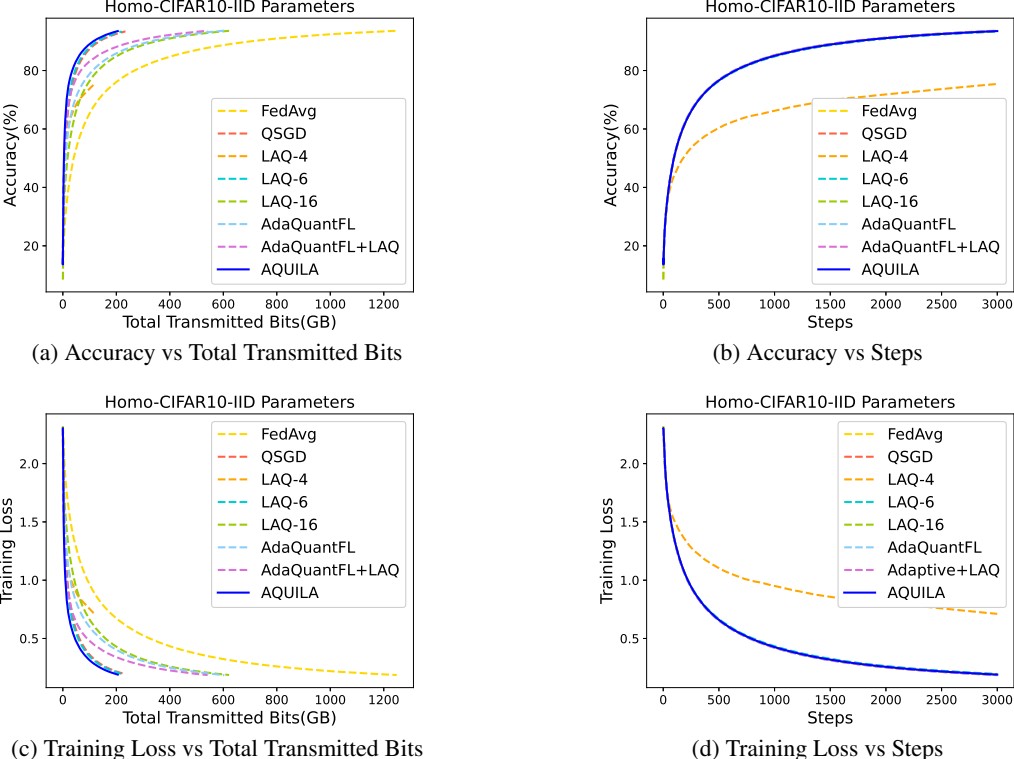

(a) Accuracy vs Total Transmitted Bits

(b) Accuracy vs Steps

(c) Training Loss vs Total Transmitted Bits

(d) Training Loss vs Steps

Figure 7: Homo-CIFAR-IID

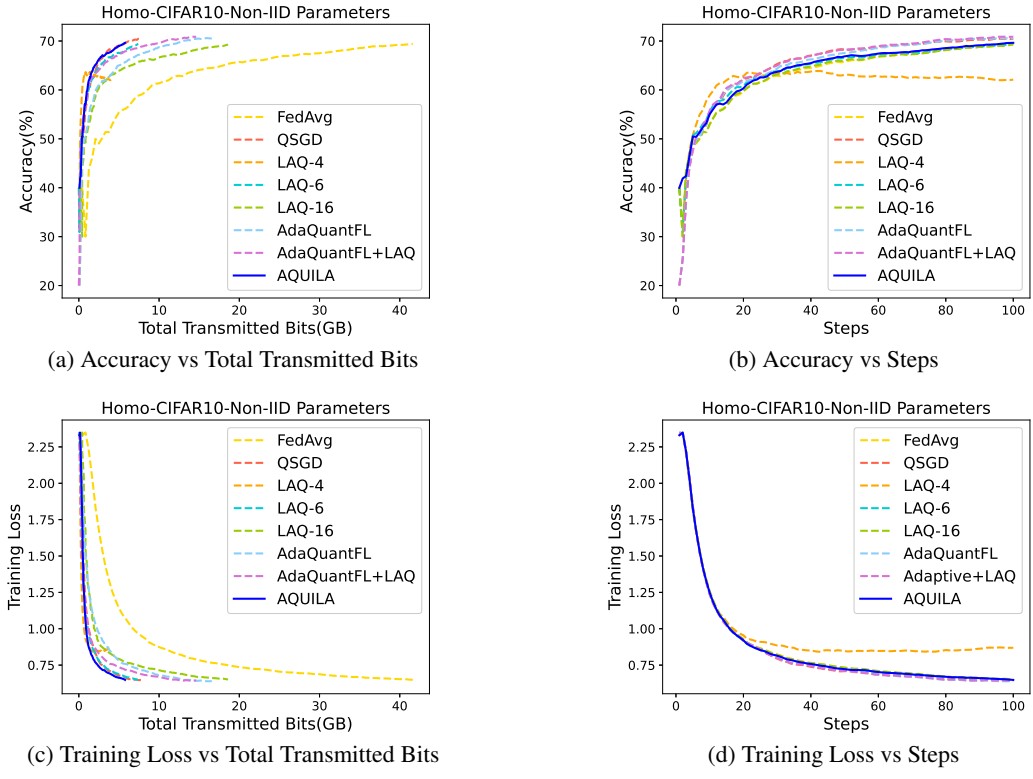

Figure 8: Homo-CIFAR-Non-IID

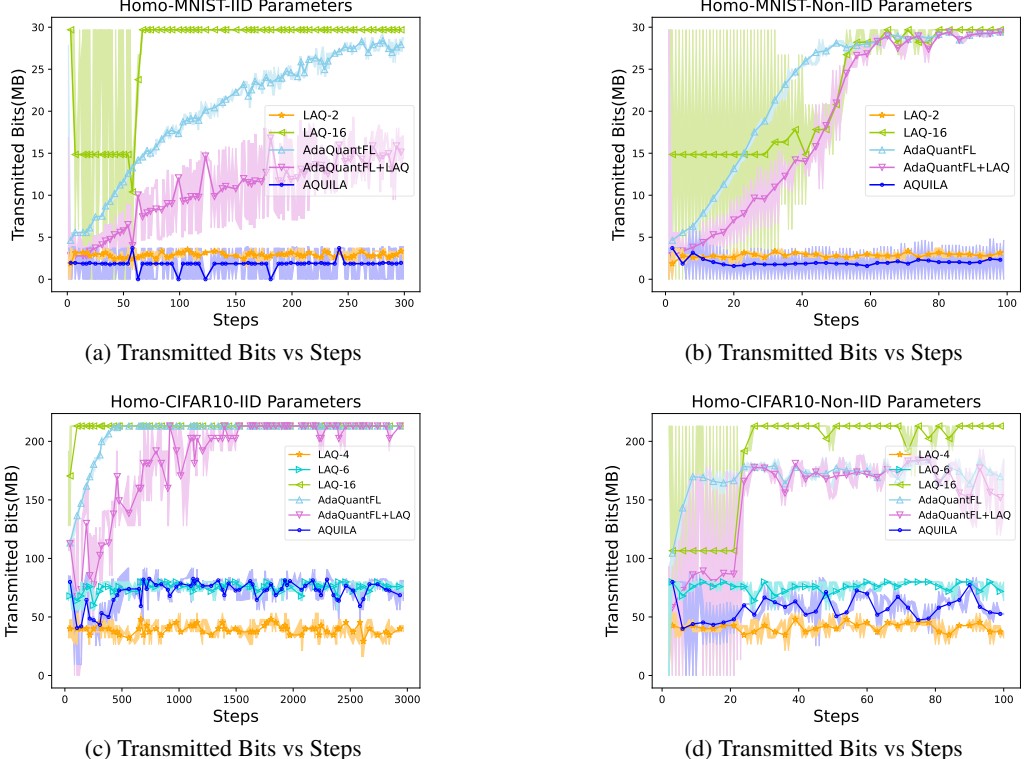

Figure 9: Homo-Transmitted Bits vs Steps

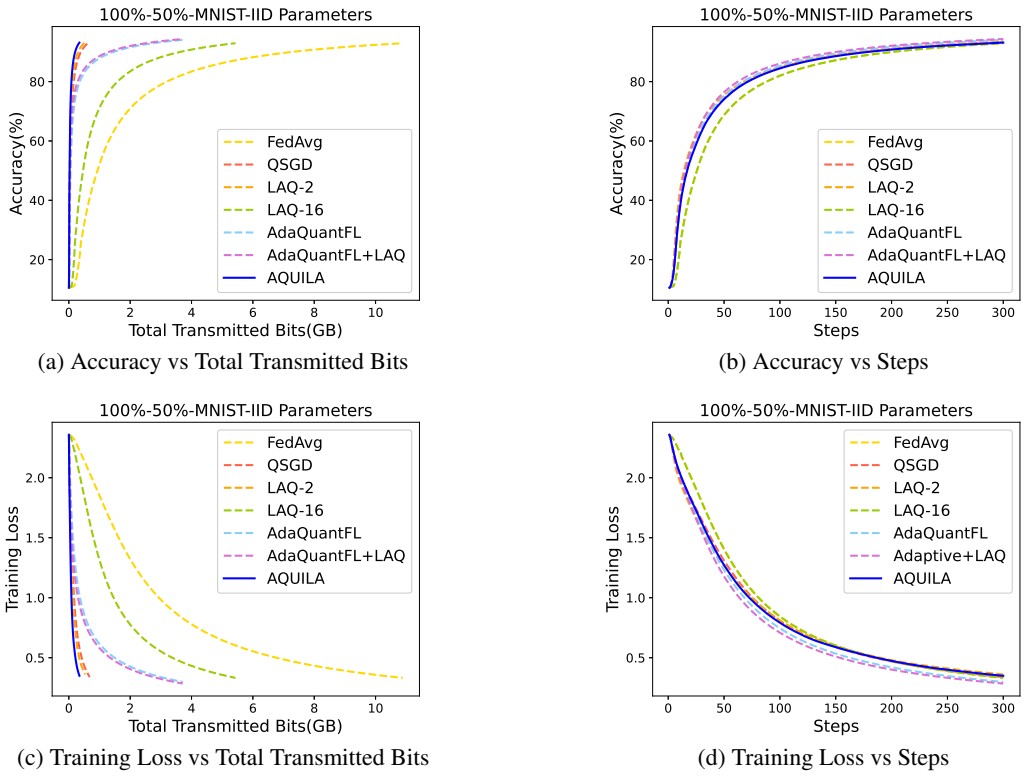

(a) Accuracy vs Total Transmitted Bits

(b) Accuracy vs Steps

(c) Training Loss vs Total Transmitted Bits

(d) Training Loss vs Steps

Figure 10: 100%-50%-MNIST-IID

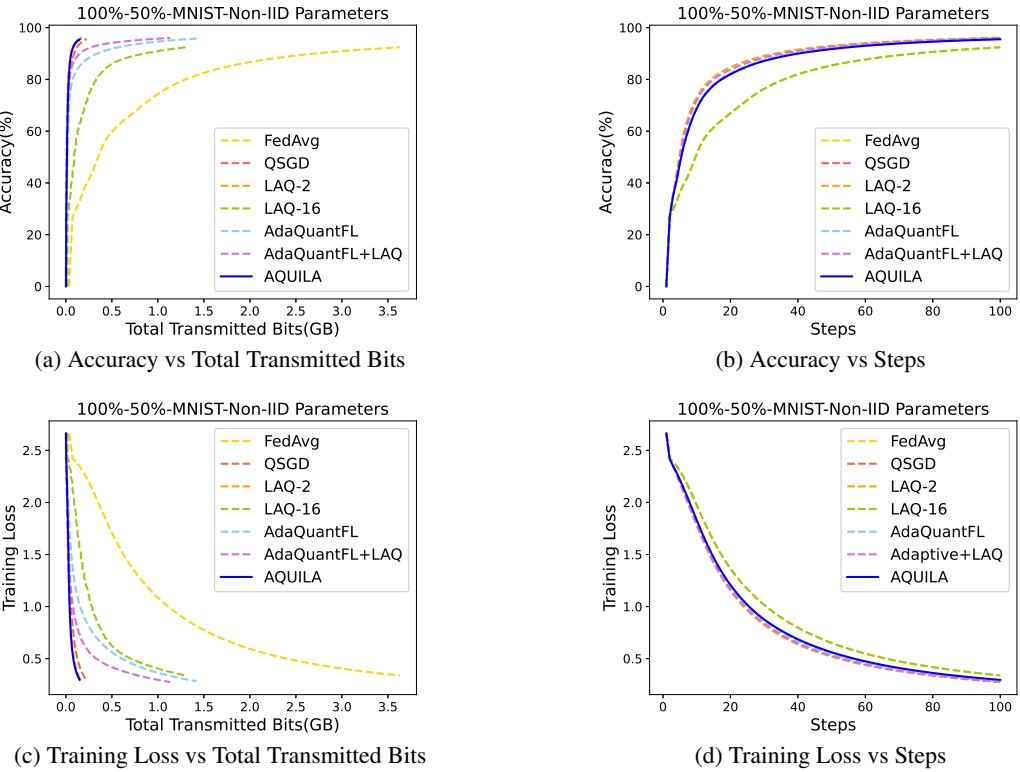

(a) Accuracy vs Total Transmitted Bits

(b) Accuracy vs Steps

(c) Training Loss vs Total Transmitted Bits

(d) Training Loss vs Steps

Figure 11: 100%-50%-MNIST-Non-IID

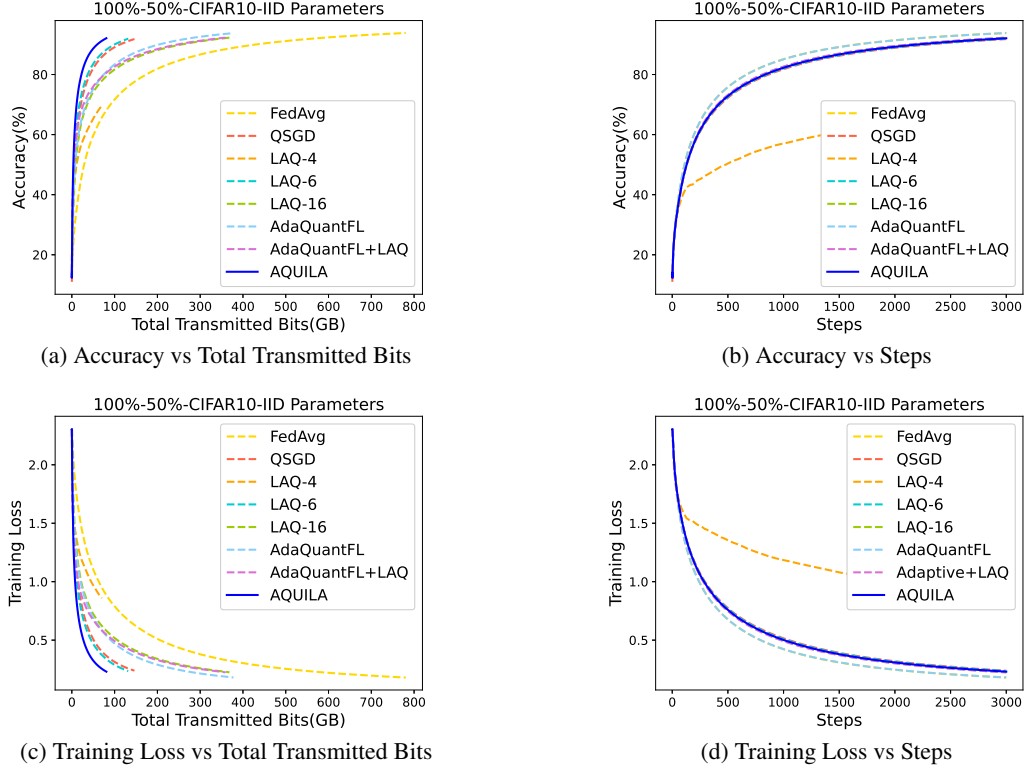

(a) Accuracy vs Total Transmitted Bits

(b) Accuracy vs Steps

(c) Training Loss vs Total Transmitted Bits

(d) Training Loss vs Steps

Figure 12: 100%-50%-CIFAR-IID

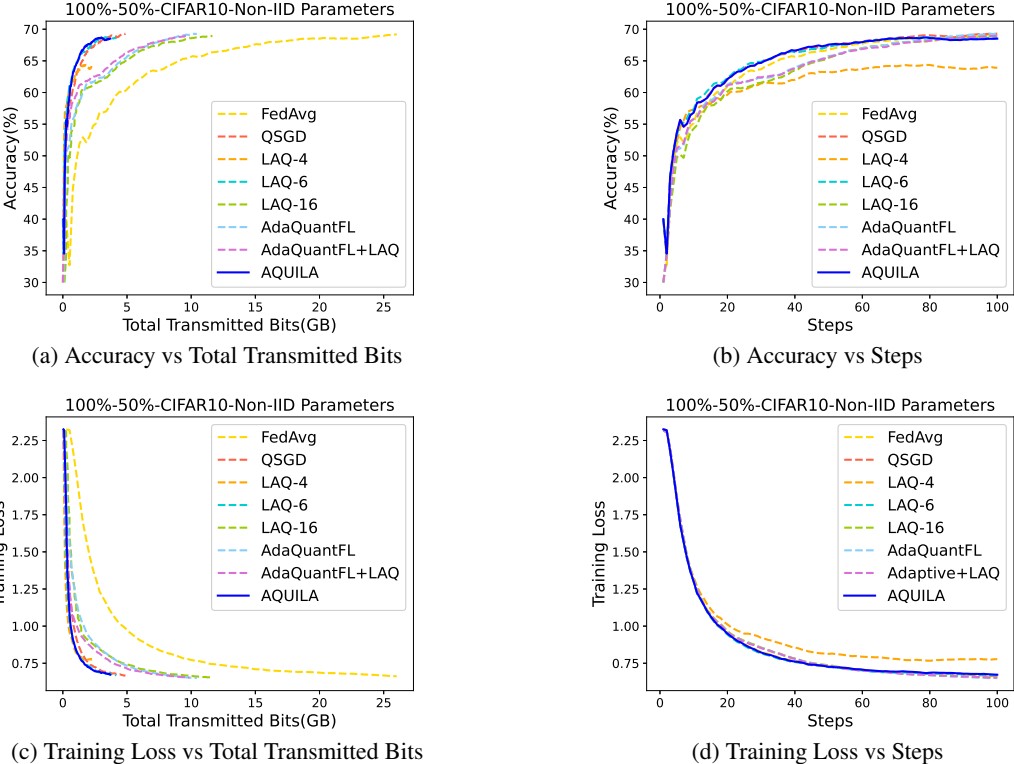

(a) Accuracy vs Total Transmitted Bits

(b) Accuracy vs Steps

(c) Training Loss vs Total Transmitted Bits

(d) Training Loss vs Steps

Figure 13: 100%-50%-CIFAR-Non-IID

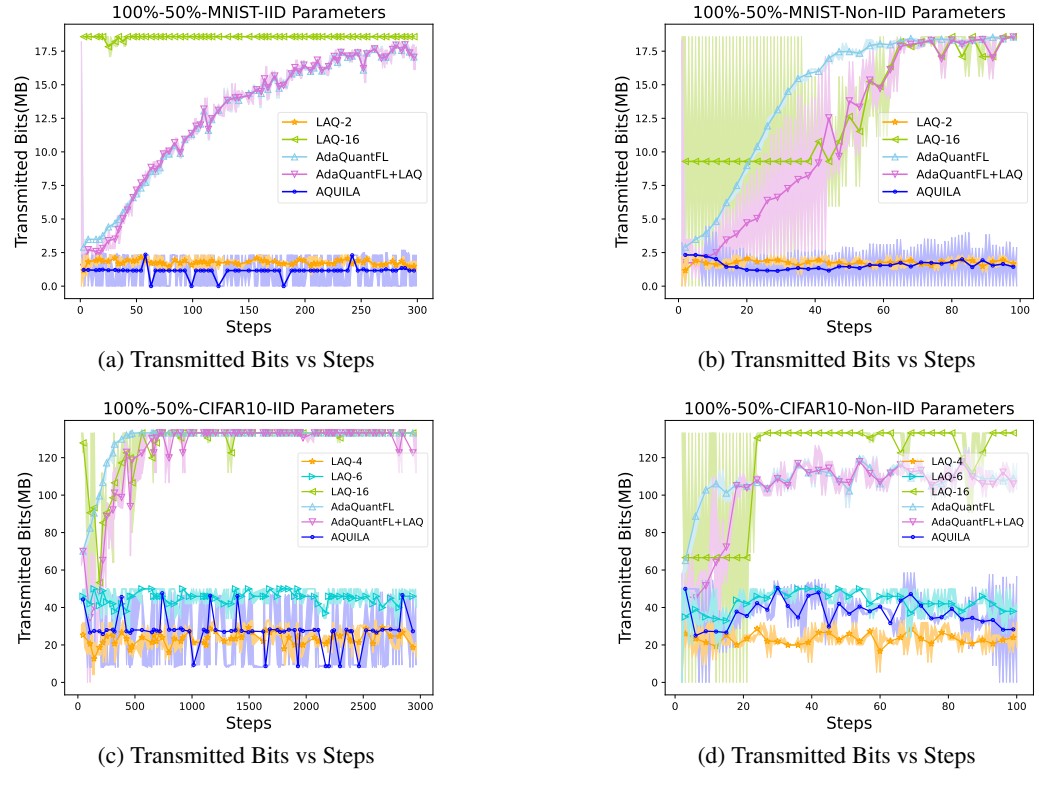

Figure 14: 100%-50%-Transmitted Bits vs Steps

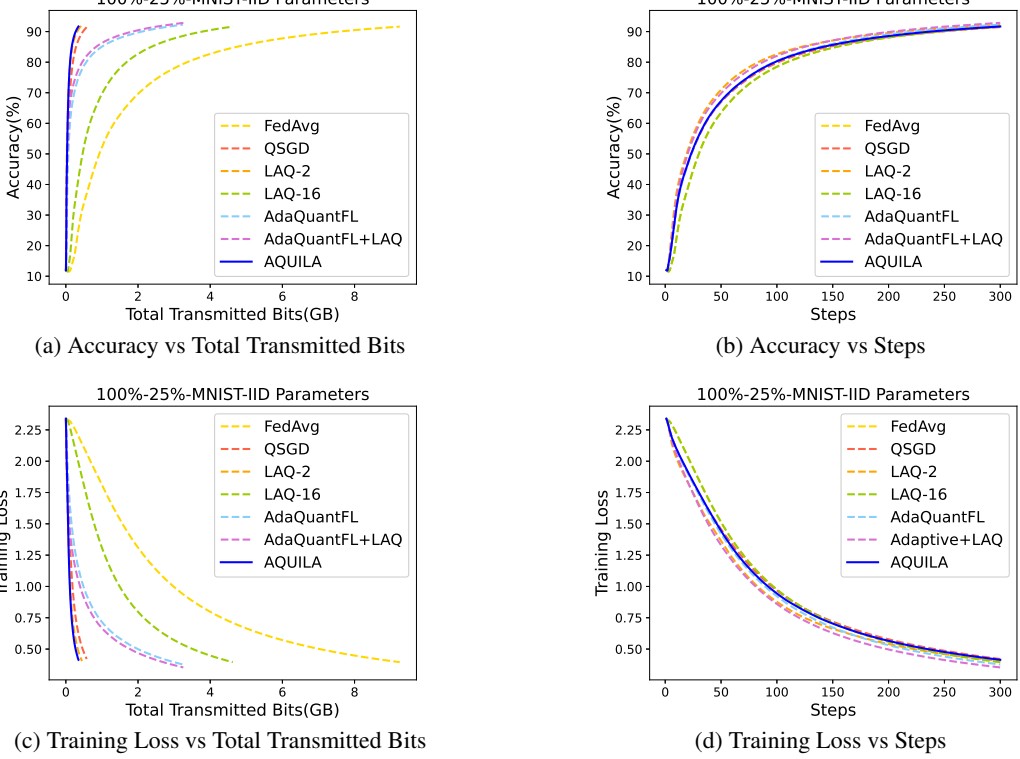

Figure 15: 100%-25%-MNIST-IID

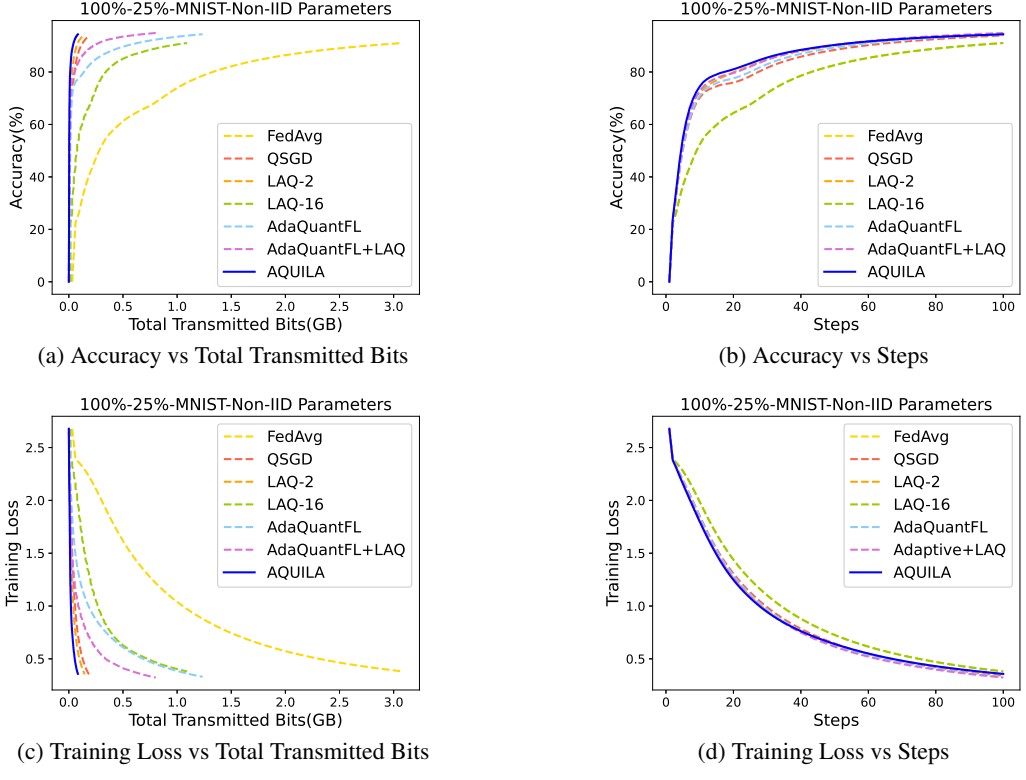

(a) Accuracy vs Total Transmitted Bits

(b) Accuracy vs Steps

(c) Training Loss vs Total Transmitted Bits

(d) Training Loss vs Steps

Figure 16: 100%-25%-MNIST-Non-IID

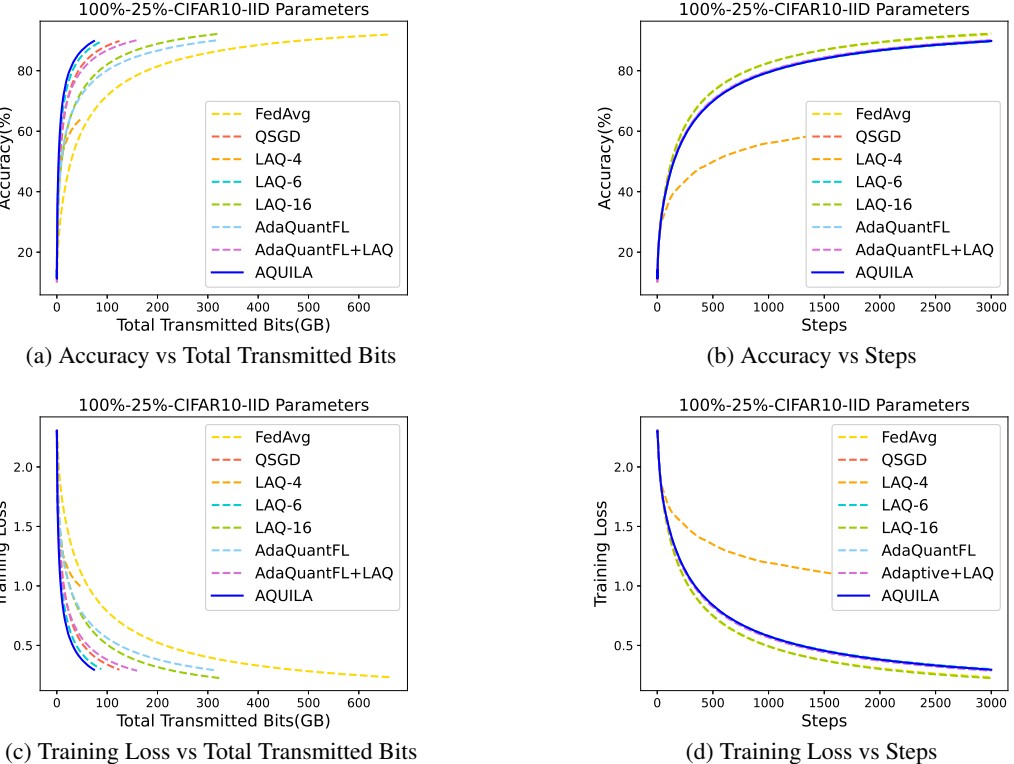

(a) Accuracy vs Total Transmitted Bits

(b) Accuracy vs Steps

(c) Training Loss vs Total Transmitted Bits

(d) Training Loss vs Steps

Figure 17: 100%-25%-CIFAR-IID

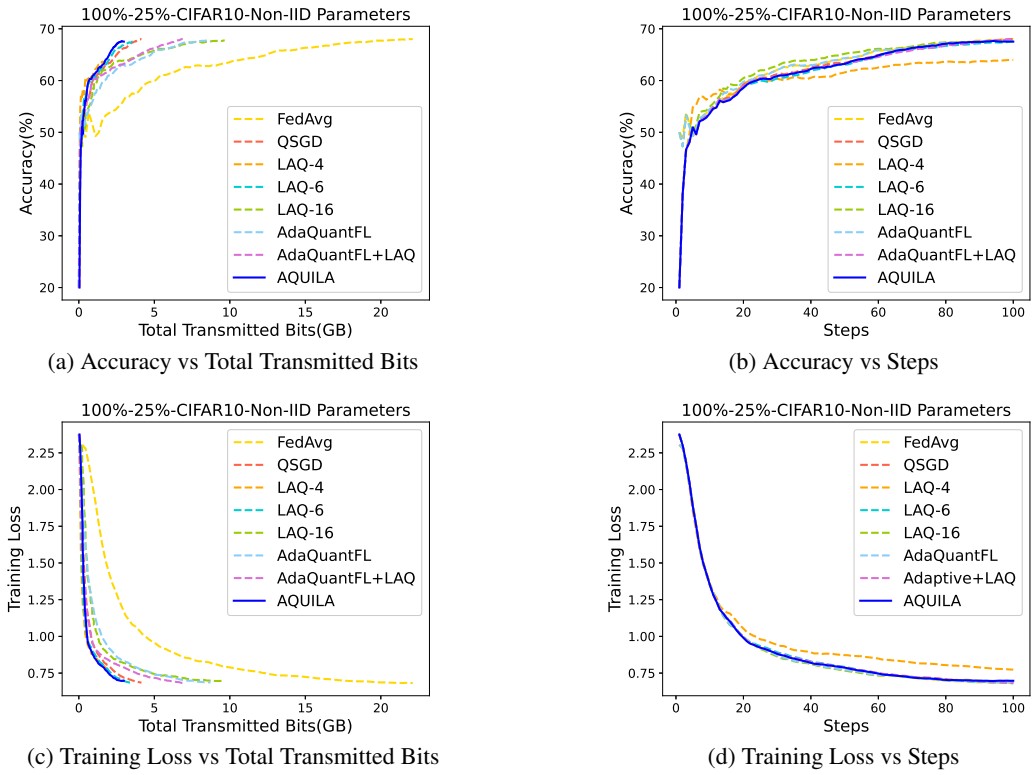

Figure 18: 100%-25%-CIFAR-Non-IID

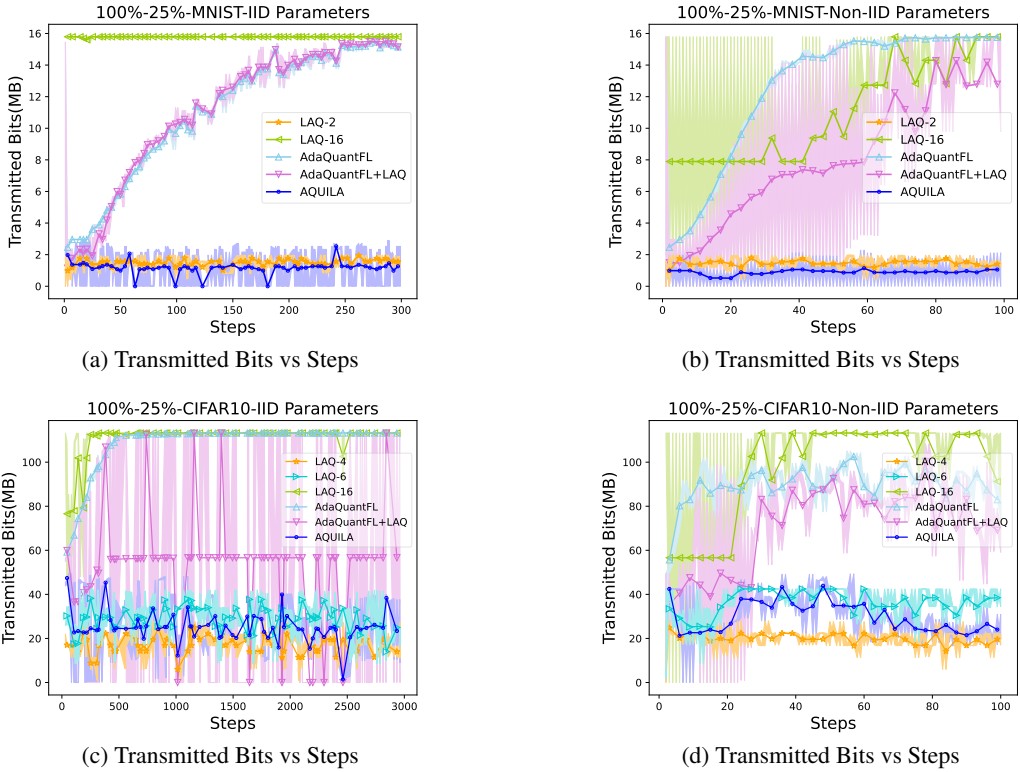

Figure 19: 100%-25%-Transmitted Bits vs Steps

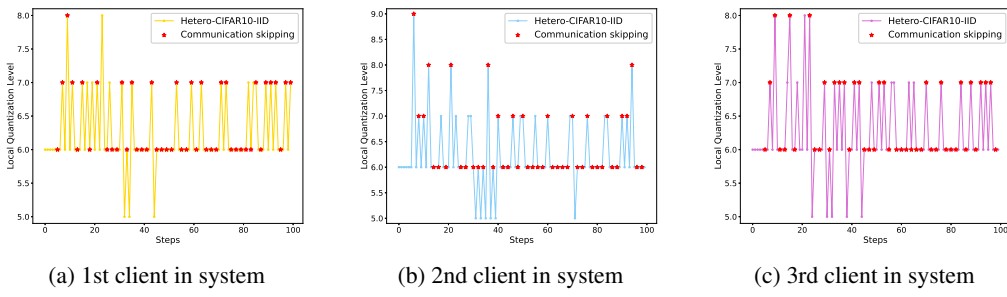

| (a) 1st client in system | (b) 2nd client in system | (c) 3rd client in system |

Figure 20: The illustration about AQUILA's ability to suppress high-bit transmission. Each figure shows the relationship between local quantization level and communication skipping of one local client during training with heterogeneous models in the IID scenario with CIFAR10 dataset. These figures imply a trend that communications with relatively high quantization level (e.g. higher than the initial level) chosen by (7) are mostly skipped by (8).

## A.2 MATHEMATICAL PROOF

### A.2.1 PROOF OF THEOREM 1

Without lazy aggregation, the aggregated model at iteration $k$ should be:

$$\boldsymbol{\theta}^{k+1} = \boldsymbol{\theta}^k - \frac{\alpha}{M} \sum_{m \in \mathbb{M}} Q_{b_m^k}(\boldsymbol{g}_m^k). \tag{13}$$

With lazy aggregation, the actual aggregated model at iteration $k$ is:

$$\hat{\boldsymbol{\theta}}^{k+1} = \boldsymbol{\theta}^k - \frac{\alpha}{M} \sum_{m \in \mathbb{M} \setminus \mathbb{M}_0^k} Q_{b_m^k}(\boldsymbol{g}_m^k) - \frac{\alpha}{M} \sum_{m \in \mathbb{M}_0^k} Q_{\hat{b}_m^{k-1}}(\hat{\boldsymbol{g}}_m^{k-1}). \tag{14}$$

With (13) and (14), the gradient loss caused by skipping can be written as:

$$
\begin{aligned}
\left\| \hat{\boldsymbol{\theta}}^{k+1} - \boldsymbol{\theta}^{k+1} \right\|_2^2 &= \left\| \frac{\alpha}{M} \sum_{m \in \mathbb{M}_0^k} Q_{b_m^k}(\boldsymbol{g}_m^k) - Q_{\hat{b}_m^{k-1}}(\hat{\boldsymbol{g}}_m^{k-1}) \right\|_2^2 \\
&\leq \frac{\alpha^2 M}{M^2} \sum_{m \in \mathbb{M}} \left\| Q_{b_m^k}(\boldsymbol{g}_m^k) - Q_{\hat{b}_m^{k-1}}(\hat{\boldsymbol{g}}_m^{k-1}) \right\|_2^2 \\
&= \frac{\alpha^2}{M} \sum_{m \in \mathbb{M}} \left\| [Q_{b_m^k}(\boldsymbol{g}_m^k) - \boldsymbol{g}_m^k] - [Q_{\hat{b}_m^{k-1}}(\hat{\boldsymbol{g}}_m^{k-1}) - (\hat{\boldsymbol{g}}_m^{k-1})] + (\boldsymbol{g}_m^k - \hat{\boldsymbol{g}}_m^{k-1}) \right\|_2^2 \\
&\leq \frac{3\alpha^2}{M} \sum_{m \in \mathbb{M}} \left\| \boldsymbol{g}_m^k - \hat{\boldsymbol{g}}_m^{k-1} \right\|_2^2 + \frac{3\alpha^2}{M} \sum_{m \in \mathbb{M}} \left( \left\| \varepsilon_{b_m^k}(\boldsymbol{g}_m^k) \right\|_2^2 + \left\| \varepsilon_{\hat{b}_m^{k-1}}(\hat{\boldsymbol{g}}_m^{k-1}) \right\|_2^2 \right),
\end{aligned} \tag{15}
$$

where the two inequalities follow from: $\left\| \sum_{i=1}^n a_i \right\|_2^2 \leq n \sum_{i=1}^n \|a_i\|_2^2$.

Take expectation of both sides of (15):

$$
\begin{aligned}
\mathbb{E}[\left\| \hat{\boldsymbol{\theta}}^{k+1} - \boldsymbol{\theta}^{k+1} \right\|_2^2] &\leq \frac{3\alpha^2}{M} \sum_{m \in \mathbb{M}} \mathbb{E}[\left\| \boldsymbol{g}_m^k - \hat{\boldsymbol{g}}_m^{k-1} \right\|_2^2] + \frac{3\alpha^2}{M} \sum_{m \in \mathbb{M}} (\mathbb{E}[\left\| \varepsilon_{b_m^k}(\boldsymbol{g}_m^k) \right\|_2^2] + \mathbb{E}[\left\| \varepsilon_{\hat{b}_m^{k-1}}(\hat{\boldsymbol{g}}_m^{k-1}) \right\|_2^2]) \\
&\leq \frac{3\alpha^2}{M} \sum_{m \in \mathbb{M}} \mathbb{E}[\left\| \boldsymbol{g}_m^k - \hat{\boldsymbol{g}}_m^{k-1} \right\|_2^2] + \frac{3\alpha^2}{M} \sum_{m \in \mathbb{M}} \left( \frac{d\sigma^2}{(\hat{b}_m^{k-1})^2} + \frac{d\sigma^2}{(b_m^k)^2} \right) \\
&\leq \frac{3\alpha^2}{M} \sum_{m \in \mathbb{M}} \frac{d_m}{K} \left\| \boldsymbol{g}_m^k - \hat{\boldsymbol{g}}_m^{k-1} \right\|_2^2 + \frac{3\alpha^2}{M} \sum_{m \in \mathbb{M}} \left( \frac{d\sigma^2}{(\hat{b}_m^{k-1})^2} + \frac{d\sigma^2}{(b_m^k)^2} \right),
\end{aligned} \tag{16}
$$

where the second inequality follows from Assumption 2 with $\sigma^2$ denoting the maximum $\|w\|_2^2$ and $q_b = d/b^2$, and the last inequality is resulted from the conclusion in LAQ (Sun et al., 2020), which indicates that the client $m$ communicates with the server at most $d_m$ rounds in total $K$ iterations, where $d_m$ is a constant related to $L_m$.

With the definition of $B_m$ and $C_m^k = d \left\lceil \log_2(b_m^k + 1) \right\rceil + d + 32$, the expected gradient loss can be written as:

$$
\begin{aligned}
\mathbb{E}[\left\|\hat{\boldsymbol{\theta}}^{k+1} - \boldsymbol{\theta}^{k+1}\right\|_2^2] &\leq \frac{3\alpha^2}{M} \sum_{m \in \mathbb{M}} \frac{d_m C_m^k}{B_m} \left\|\boldsymbol{g}_m^k - \hat{\boldsymbol{g}}_m^{k-1}\right\|_2^2 + \frac{3\alpha^2}{M} \sum_{m \in \mathbb{M}} (\frac{d\sigma^2}{(\hat{b}_m^{k-1})^2} + \frac{d\sigma^2}{(b_m^k)^2}) \\
&= \frac{3\alpha^2}{M} \sum_{m \in \mathbb{M}} \frac{d_m d \left\lceil \log_2(b_m^k + 1) \right\rceil}{B_m} \left\|\boldsymbol{g}_m^k - \hat{\boldsymbol{g}}_m^{k-1}\right\|_2^2 + \frac{3\alpha^2}{M} \sum_{m \in \mathbb{M}} (\frac{d\sigma^2}{(\hat{b}_m^{k-1})^2} + \frac{d\sigma^2}{(b_m^k)^2}) \\
&\quad + \frac{3\alpha^2}{M} \sum_{m \in \mathbb{M}} \frac{d_m(d + 32)}{B_m} \left\|\boldsymbol{g}_m^k - \hat{\boldsymbol{g}}_m^{k-1}\right\|_2^2 \\
&\leq \frac{3\alpha^2}{M} \sum_{m \in \mathbb{M}} \frac{d_m d \left\lceil \log_2(4b_m^k) \right\rceil}{B_m} \left\|\boldsymbol{g}_m^k - \hat{\boldsymbol{g}}_m^{k-1}\right\|_2^2 + \frac{3\alpha^2}{M} \sum_{m \in \mathbb{M}} (\frac{d\sigma^2}{(\hat{b}_m^{k-1})^2} + \frac{d\sigma^2}{(b_m^k)^2}) \\
&\quad + \frac{3\alpha^2}{M} \sum_{m \in \mathbb{M}} \frac{d_m(d + 32)}{B_m} \left\|\boldsymbol{g}_m^k - \hat{\boldsymbol{g}}_m^{k-1}\right\|_2^2.
\end{aligned}
\tag{17}
$$

This completes the proof for Theorem 1.

### A.2.2 DERIVATION DETAILS OF ADAPTIVE QUANTIZATION CRITERION (7)

Let $H$ be the upper bound to be minimized:

$$
\begin{aligned}
H &= \frac{3\alpha^2}{M} \sum_{m \in \mathbb{M}} \frac{d_m d \left\lceil \log_2(4b_m^k) \right\rceil}{B_m} \left\|\boldsymbol{g}_m^k - \hat{\boldsymbol{g}}_m^{k-1}\right\|_2^2 + \frac{3\alpha^2}{M} \sum_{m \in \mathbb{M}} (\frac{d\sigma^2}{(\hat{b}_m^{k-1})^2} + \frac{d\sigma^2}{(b_m^k)^2}) \\
&\quad + \frac{3\alpha^2}{M} \sum_{m \in \mathbb{M}} \frac{d_m(d + 32)}{B_m} \left\|\boldsymbol{g}_m^k - \hat{\boldsymbol{g}}_m^{k-1}\right\|_2^2.
\end{aligned}
\tag{18}
$$

The first derivative is:

$$
\frac{\partial H}{\partial b_m^k} = \frac{3\alpha^2 d_i d \left\|\boldsymbol{g}_m^k - \hat{\boldsymbol{g}}_m^{k-1}\right\|_2^2 \left\lceil \log_2 e \right\rceil}{B_m M b_m^k} - \frac{6\alpha^2 d\sigma^2}{M(b_m^k)^3}.
\tag{19}
$$

Let $\frac{\partial H}{\partial b_m^k} = 0$, there is:

$$
(b_m^k)^* = \sqrt{\frac{2\sigma^2 B_m log_e(2)}{d_i \left\|\boldsymbol{g}_m^k - \hat{\boldsymbol{g}}_m^{k-1}\right\|_2^2}}.
\tag{20}
$$

Dividing $(b_m^k)^*$ by $(b_m^1)$:

$$
(b_m^k)^* = b_m^1 \sqrt{\frac{\left\|\boldsymbol{g}_m^1 - \hat{\boldsymbol{g}}_m^0\right\|_2^2}{\left\|\boldsymbol{g}_m^k - \hat{\boldsymbol{g}}_m^{k-1}\right\|_2^2}}.
\tag{21}
$$

For iteration $k = 1$, we set $b_m^1$ for each client $m$ as the intial quantization level $b_0$, and thus we get the adaptive quantization criterion (7) for iteration $k = 1, 2, ..., K$:

$$
(b_m^k)^* = b_0 \sqrt{\frac{\left\|\boldsymbol{g}_m^1 - \hat{\boldsymbol{g}}_m^0\right\|_2^2}{\left\|\boldsymbol{g}_m^k - \hat{\boldsymbol{g}}_m^{k-1}\right\|_2^2}}.
\tag{22}
$$

### A.2.3 PROOF OF THEOREM 2

With the adjusted communication criterion (8), all transmitted gradient updates are larger than a threshold after quantization. Therefore, all transmitted gradient updates satisfy:

$$
\begin{aligned}
\left\| \boldsymbol{g}_m^k - \hat{\boldsymbol{g}}_m^{k-1} \right\|_2^2 &= \left\| \boldsymbol{g}_m^k - Q_{b_m^k}\left(\boldsymbol{g}_m^k\right) + Q_{\hat{b}_m^{k-1}}\left(\hat{\boldsymbol{g}}_m^{k-1}\right) - \hat{\boldsymbol{g}}_m^{k-1} + Q_{b_m^k}\left(\boldsymbol{g}_m^k\right) - Q_{\hat{b}_m^{k-1}}\left(\hat{\boldsymbol{g}}_m^{k-1}\right) \right\|_2^2 \\
&\approx \left\| Q_{b_m^k}\left(\boldsymbol{g}_m^k\right) - Q_{\hat{b}_m^{k-1}}\left(\hat{\boldsymbol{g}}_m^{k-1}\right) \right\|_2^2 \\
&\geq \frac{\sum_{d=1}^D \xi_d \left\| \boldsymbol{\theta}^{k+1-d} - \boldsymbol{\theta}^{k-d} \right\|_2^2}{\alpha^2 M^2} + 3\left( \left\| \varepsilon_{b_m^k}(\boldsymbol{g}_m^k) \right\|_2^2 + \left\| \varepsilon_{\hat{b}_m^{k-1}}(\hat{\boldsymbol{g}}_m^{k-1}) \right\|_2^2 \right).
\end{aligned} \tag{23}
$$

then,

$$
\begin{aligned}
(b_m^k)^* &= b_0 \sqrt{\frac{\left\| \boldsymbol{g}_m^1 - \hat{\boldsymbol{g}}_m^0 \right\|_2^2}{\left\| \boldsymbol{g}_m^k - \hat{\boldsymbol{g}}_m^{k-1} \right\|_2^2}} \\
&\leq b_0 \sqrt{\frac{\left\| \boldsymbol{g}_m^1 - \hat{\boldsymbol{g}}_m^0 \right\|_2^2}{\frac{1}{\alpha^2 M^2} \sum_{d=1}^D \xi_d \left\| \boldsymbol{\theta}^{k+1-d} - \boldsymbol{\theta}^{k-d} \right\|_2^2 + 3\left( \left\| \varepsilon_{b_m^k}(\boldsymbol{g}_m^k) \right\|_2^2 + \left\| \varepsilon_{\hat{b}_m^{k-1}}(\hat{\boldsymbol{g}}_m^{k-1}) \right\|_2^2 \right)}} \\
&\leq b_0 \sqrt{\frac{\left\| \boldsymbol{g}_m^1 - \hat{\boldsymbol{g}}_m^0 \right\|_2^2}{\frac{2}{\alpha^2 M^2 L} \sum_{d=1}^D \xi_d (f(\boldsymbol{\theta}^{k+1-d}) - f(\boldsymbol{\theta}^{k-d})) + 3\left( \left\| \varepsilon_{b_m^k}(\boldsymbol{g}_m^k) \right\|_2^2 + \left\| \varepsilon_{\hat{b}_m^{k-1}}(\hat{\boldsymbol{g}}_m^{k-1}) \right\|_2^2 \right)}} \\
&\leq b_0 \sqrt{\frac{\left\| \boldsymbol{g}_m^1 - \hat{\boldsymbol{g}}_m^0 \right\|_2^2}{\frac{2\xi_d}{\alpha^2 M^2 L}[f(\boldsymbol{\theta}^k) - f(\boldsymbol{\theta}^{k-D})] + 3\left( \left\| \varepsilon_{b_m^k}(\boldsymbol{g}_m^k) \right\|_2^2 + \left\| \varepsilon_{\hat{b}_m^{k-1}}(\hat{\boldsymbol{g}}_m^{k-1}) \right\|_2^2 \right)}} \\
&\leq b_0 \sqrt{\frac{\left\| \boldsymbol{g}_m^1 - \hat{\boldsymbol{g}}_m^0 \right\|_2^2}{\frac{2\xi_d}{\alpha^2 M^2 L}[f(\boldsymbol{\theta}^k) - f(\boldsymbol{\theta}^*)]}} \\
&\leq b_0 \sqrt{\frac{(L_m)^2 \left\| \boldsymbol{\theta}^1 - \boldsymbol{\theta}^0 \right\|_2^2}{\frac{2\xi_d}{\alpha^2 M^2 L}[f(\boldsymbol{\theta}^k) - f(\boldsymbol{\theta}^*)]}},
\end{aligned} \tag{24}
$$

where the second inequality comes from Assumption 1. Let $P = \dfrac{f(\boldsymbol{\theta}^0) - f(\boldsymbol{\theta}^*)}{L \left\| \boldsymbol{\theta}^1 - \boldsymbol{\theta}^0 \right\|_2^2}$. Therefore, if $\alpha \leq (1/ML)\sqrt{2P\xi/L}$ holds, then in the experiment setting where $\xi_1 = \xi_2 = ... = \xi_D = \xi$ and $L \approx L_m$ for all $m \in \mathbb{M}$, there is:

$$
(b_m^k)^* \leq b_0 \sqrt{\frac{(L_m)^2 \left\| \boldsymbol{\theta}^1 - \boldsymbol{\theta}^0 \right\|_2^2}{\frac{2\xi_d}{\alpha^2 M^2 L}[f(\boldsymbol{\theta}^k) - f(\boldsymbol{\theta}^*)]}} \leq b_0 \sqrt{\frac{f(\boldsymbol{\theta}^0) - f(\boldsymbol{\theta}^*)}{f(\boldsymbol{\theta}^k) - f(\boldsymbol{\theta}^*)}}. \tag{25}
$$

### A.2.4 DEVELOPMENT OF LAZY AGGREGATION CRITERION (8)

$$
\left\| Q_{b_m^k} \left( \boldsymbol{g}_m^k \right) - Q_{\hat{b}_m^{k-1}} \left( \hat{\boldsymbol{g}}_m^{k-1} \right) \right\|_2^2
$$

$$
= \left\| Q_{b_m^k} \left( \boldsymbol{g}_m^k \right) - \boldsymbol{g}_m^k - Q_{\hat{b}_m^{k-1}} \left( \hat{\boldsymbol{g}}_m^{k-1} \right) + \hat{\boldsymbol{g}}_m^{k-1} + \boldsymbol{g}_m^k - \hat{\boldsymbol{g}}_m^{k-1} \right\|_2^2
$$

$$
\leq 2 \left\| \boldsymbol{g}_m^k - \hat{\boldsymbol{g}}_m^{k-1} \right\|_2^2 + 2 \left\| \varepsilon_{b_m^k} (\boldsymbol{g}_m^k) - \varepsilon_{\hat{b}_m^{k-1}}(\hat{\boldsymbol{g}}_m^{k-1}) \right\|_2^2
$$

$$
\leq 2 L_m^2 \left\| \boldsymbol{\theta}^k - \boldsymbol{\theta}^{k-d'} \right\|_2^2 + 2 \left\| \varepsilon_{b_m^k} (\boldsymbol{g}_m^k) - \varepsilon_{\hat{b}_m^{k-1}}(\hat{\boldsymbol{g}}_m^{k-1}) \right\|_2^2
$$

$$
= 2 L_m^2 \left\| \sum_{d=1}^{d'} \boldsymbol{\theta}^{k+1-d} - \boldsymbol{\theta}^{k-d} \right\|_2^2 + 2 \left\| \varepsilon_{b_m^k} (\boldsymbol{g}_m^k) - \varepsilon_{\hat{b}_m^{k-1}}(\hat{\boldsymbol{g}}_m^{k-1}) \right\|_2^2
$$

$$
\leq 2 L_m^2 d' \sum_{d=1}^{d'} \left\| \boldsymbol{\theta}^{k+1-d} - \boldsymbol{\theta}^{k-d} \right\|_2^2 + 2 \left\| \varepsilon_{b_m^k} (\boldsymbol{g}_m^k) - \varepsilon_{\hat{b}_m^{k-1}}(\hat{\boldsymbol{g}}_m^{k-1}) \right\|_2^2, \tag{26}
$$

where the second inequality comes from:

$$
\hat{\boldsymbol{g}}_m^{k-1} = \nabla f_m(\boldsymbol{\theta}^{k-d'}). \tag{27}
$$

$$
\boldsymbol{g}_m^k = \nabla f_m(\boldsymbol{\theta}^k). \tag{28}
$$

$$
\left\| \nabla f_m(\boldsymbol{\theta}^k) - \nabla f_m(\boldsymbol{\theta}^{k-d'}) \right\|_2^2 \leq L_m^2 \left\| \boldsymbol{\theta}^k - \boldsymbol{\theta}^{k-d'} \right\|_2^2. \tag{29}
$$

Following LAQ's definition, redefine $d_m, m \in \mathbb{M}$ as:

$$
d_m := \max_d \left\{ d | L_m^2 \leq \xi_d / (2\alpha^2 M^2 D), d \in 1, 2, ..., D \right\}. \tag{30}
$$

With the definition of $d_m$ and $\xi_1 \geq \xi_2 \geq ... \geq \xi_D$, there is:

$$
L_m^2 \leq \frac{\xi_{d'}}{2\alpha^2 M^2 D}, \text{for all } d' \text{ satisfying} 1 \leq d' \leq d_m. \tag{31}
$$

With (26) and (31), we have:

$$
\left\| Q_{b_m^k} \left( \boldsymbol{g}_m^k \right) - Q_{\hat{b}_m^{k-1}} \left( \hat{\boldsymbol{g}}_m^{k-1} \right) \right\|_2^2
$$

$$
\leq \frac{1}{\alpha^2 M^2} \sum_{d=1}^{d'} \xi_{d'} \left\| \boldsymbol{\theta}^{k+1-d} - \boldsymbol{\theta}^{k-d} \right\|_2^2 + 2 \left\| \varepsilon_{b_m^k} (\boldsymbol{g}_m^k) - \varepsilon_{\hat{b}_m^{k-1}}(\hat{\boldsymbol{g}}_m^{k-1}) \right\|_2^2
$$

$$
\leq \frac{1}{\alpha^2 M^2} \sum_{d=1}^{D} \xi_d \left\| \boldsymbol{\theta}^{k+1-d} - \boldsymbol{\theta}^{k-d} \right\|_2^2 + 2 \left\| \varepsilon_{b_m^k} (\boldsymbol{g}_m^k) - \varepsilon_{\hat{b}_m^{k-1}}(\hat{\boldsymbol{g}}_m^{k-1}) \right\|_2^2. \tag{32}
$$

