# OpenReview forum: "AQUILA: Communication Efficient Federated Learning with Adaptive Quantization of Lazily-Aggregated Gradients"
_ICLR.cc/2022/Conference — ICLR 2022 Submitted_

### Official Review · Reviewer_Dvx6 · 2021-11-02

**Correctness:** 4
**Technical Novelty And Significance:** 2
**Empirical Novelty And Significance:** 3
**Recommendation:** 6
**Confidence:** 3

**Main Review:**

**A Strengths:**
1) Experiments validate the superiority of the proposed method over each of its individual components (LAQ, AdaQuantFL,), as well as their naive combination, and other well-known FL methods.

2) At a high level, the idea itself of combining two orthogonal directions in Federated learning, transmitting infrequently and compressing the updates, is very interesting. The fact that the trivial combination of these two approaches isn't as effective and one needs to intricately design the overall scheme is surprising, and something to keep in mind for future research in this area.

**B Weaknesses:**
1)  It was disappointing not to see an explicit convergence rate of the proposed method.

2) Building on the previous point, a lack of comparison with local methods in convergence rate and experiments was disappointing. I think these comparisons are needed for a fair assessment of the paper.

3) Since this paper is mainly about combining existing ideas, there is not a lot of algorithmic novelty in the paper.

**C Presentation:**
1) The paper is well-written and easy to follow.

2) Is there any specific reason that authors choose to minimize the upper bound of the quantity on the LHS of Theorem 1. Choosing different metrics would lead to different bit allocation strategies. Why was this particular metric chosen then?

**Summary Of The Paper:**

The paper proposes combining two orthogonal algorithms -- the lazily aggregated gradient(LAQ)method and adaptive quantization (AdaQuantFL)-- to reduce communication complexity in federated learning. In particular, while LAQ focuses on the frequency of gradient updates in each iteration, AdaQuantFL focuses on the adaptive allocation of bits across iterations. This paper proposes to combine both of these approaches for better communication complexity in federated learning.

At the heart of the combining strategy is Eq. 7, which allocates the bits used in each iteration. This equation is intern based on minimizing the upper bound on the expected deviation in the path of the iterate due to gradient skipping. The authors theoretically show that such a bit allocation strategy leads to reduced communication compared to the bit allocation strategy of AdaQuantFl.



**Summary Of The Review:**

Overall, primarily based on the experiments, I recommend the paper be accepted. However, I cannot be as confident of the paper's contribution as I would like to be because of a lack of comparison with a crucial part of the literature (B.1, B.2).

---

> ### Author Response · Authors · 2021-11-18
> **Response to Reviewer Dvx6**
>
> Thank you for your valuable feedback!
>
> **Q1: It was disappointing not to see an explicit convergence rate of the proposed method.**
>
> **Q2: Building on the previous point, a lack of comparison with local methods in convergence rate and experiments was disappointing. I think these comparisons are needed for a fair assessment of the paper.**
>
> **Response to 1 \& 2:**  Thank you for your valuable comments. Actually we had compared the convergence rate of the proposed method with local methods, which are shown as the ***accuracy vs steps*** figures and ***training loss vs steps*** figures in the Appendix.
>
> **Q3: Since this paper is mainly about combining existing ideas, there is not a lot of algorithmic novelty in the paper.**
>
> **Response:** Thank you for the professional review. We would like to highlight the novel contributions of our work as bifold:
> 1) Characterizing the expected deviation introduced by both communication frequency and quantization is a new problem. The upperbound we derive for the expected deviation is novel, which provides a deeper insight into the FL framework with both lazy aggregation and adaptive quantization.
> 2) By innovatively formulating the target problem as an optimization problem minimizing the aforementioned upperbound, we obtain a new algorithm with adaptive quantization. The main novelty of our algorithm lies in adaptive quantization criterion’s ability to cooperate with lazy aggregation strategy for higher communication efficiency while maintaining or even improving the convergence properties with low-bit transmission.
>
> We have conducted extensive experiments covering multiple non-homogeneous settings to show how the two mutually dependent factors are adjusted synergistically in AQUILA and evaluate its effectiveness in improving FL’s communication efficiency. All the experimental results demonstrate that AQUILA is highly effective (can reduce overall transmission overhead by around 50% compared to existing methods) and compatible to existing FL settings. Given the above contributions, we would like to conclude the proposed AQUILA as an innovation for both theory and application.
>
> **Q4: Is there any specific reason that authors choose to minimize the upper bound of the quantity on the LHS of Theorem 1. Choosing different metrics would lead to different bit allocation strategies. Why was this particular metric chosen then?**
>
> **Response:** Thank you for the professional review. The reason why we choose to minimize the upper bound of the quantity is:
> 1) Quantifying and minimizing the exact error is not tractable. Therefore, similar to existing work [R1], we characterize a novel upperbound and minimize it with respect to transmitted bits.
> 2) The bit allocation strategy given by the metric we select is able to exploit the interaction between the adjustment of the transmission frequency and quantization level. Specifically, it depends on gradient changes between updates similar to lazy aggregation strategy. This key property bridges the gap between the two degrees of freedom, and therefore allows us to intricately leverage these two complementary yet mutually-dependent factors for further optimizing communication efficiency in FL. The aforementioned superiorities have been proved with our experimental results.
>
> *[R1] Divyansh Jhunjhunwala, Advait Gadhikar, Gauri Joshi, and Yonina C Eldar. Adaptive quantization of model updates for communication-efficient federated learning. ICASSP, 2021.*

---

> > ### Comment · Reviewer_Dvx6 · 2021-11-29
> > **Response to the rebuttal**
> >
> > I thank the authors for their clarification. After going through the other reviews and the author's response, my review remains unchanged.

---

### Official Review · Reviewer_E4L8 · 2021-11-02

**Correctness:** 4
**Technical Novelty And Significance:** 3
**Empirical Novelty And Significance:** 3
**Recommendation:** 8
**Confidence:** 3

**Main Review:**

The idea of this paper is to develop the framework fro FL that allows to ruduce communication cost.
Authors uses the following ideas as a core: 1)  the frequency of communication rounds depends on the local gradient change (Local Gradient Aggregation); 2) adaptive quantization correlates with the logic of the Lazy Gradient Aggregation - the quantization level is smaller if the local gradient difference between the current gradient and last sent one is bigger.
The combination of these two ideas allows to reduce the communication round amount in the late iterations of algorithm comparing to AdaQuantFL.

My main concern is the following: since the server updates the model with some new quantized gradients and with old ones for the machines that have decided to skip their update we save some amount of communication rounds TO the server. However, the update of the global model is a combination of quantized updates it can be quite ``dense'' and hard to send, but on every round of the algorithm server broadcast the new model to all machines that seems to be costly. May be the broadcast should be made in the same manner as the lazy aggregation procedure?

Also I have a minor comment to the notation $\theta^{\star}$ that seems to denote $f(\theta^\star}$.

According to the experimental section, I failed to follow the plots: they are quite small on A4 paper; they are not colorblind friendly and I am colorblind. However, I still have a question to the Figures representing "transmitted bits" do they assume the communication TO server  only or together with the broadcasting.





**Summary Of The Paper:**

In this paper, authors propose the framework for federated learning that adaptively adjusts the frequency of the communication rounds and the quantization efficiency in a synergistic way.

**Summary Of The Review:**

In conclusion, I think that authors of this work made a good job. The idea of adaptation of quantization and frequency of communications is very nice; however the broadcasting procedure seems to be very time-consuming.
For me the Experimental section was impossible to follow, but I think that minor modifications of the Figures can resolve this problem.

All in all, I think that after some changes this paper can be accepted and the presented algorithm can be used in real-world FL applications.

---

> ### Author Response · Authors · 2021-11-18
> **Response to Reviewer E4L8**
>
> Thank you very much for your review and the positive reception of our work.
>
> **Q1: Since the server updates the model with some new quantized gradients and with old ones for the machines that have decided to skip their update we save some amount of communication rounds TO the server. However, the update of the global model is a combination of quantized updates it can be quite ``dense'' and hard to send, but on every round of the algorithm server broadcast the new model to all machines that seems to be costly. May be the broadcast should be made in the same manner as the lazy aggregation procedure?**
>
> **Response:** Thanks for the professional review. We would like to clarify that research on mitigating communication overhead in FL primarily focuses on communications to the server [R1-R3], since the clients/devices usually have limited bandwidth and slow network connections compared to the central server. However, we acknowledge that saving the broadcast cost with techniques like AQUILA would be an interesting topic for future
> research, which has been taken into consideration in our next work.
>
> *[R1] Wei Wen, Cong Xu, Feng Yan, Chunpeng Wu, Yandan Wang, Yiran Chen, and Hai Li. Terngrad: Ternary gradients to reduce communication in distributed deep learning. NeurIPS, 2017.*
>
> *[R2] Tianyi Chen, Georgios B Giannakis, Tao Sun, and Wotao Yin. LAG: Lazily aggregated gradient for communication-efficient distributed learning. NeurIPS, 2018.*
>
> *[R3] Divyansh Jhunjhunwala, Advait Gadhikar, Gauri Joshi, and Yonina C Eldar. Adaptive quantization of model updates for communication-efficient federated learning. ICASSP, 2021.*
>
> **Q2: Also I have a minor comment to the notation $\theta^*$ that seems to denote $f(\theta^{*})$.**
>
> **Response:** Thank you for pointing out the confusion. In this paper, we denote the optimal model parameters by $\theta^*$, and the corresponding objective value is referred to as $f(\theta^*)$. We have carefully revised the manuscript and corrected some typos that caused the misunderstanding. Thank you again for the careful review.
>
> **Q3: According to the experimental section, I failed to follow the plots: they are quite small on $A4$ paper; they are not colorblind friendly and I am colorblind. However, I still have a question to the Figures representing "transmitted bits". Do they assume the communication TO server only or together with the broadcasting.**
>
> **Response:** Thank you for the constructive comments. We have to apologize that we did not consider the problem of colorblindness while plotting our figures, and we have made the following improvements to the figures in the experiment part:
> 1) in Figure 3, all lines except AQUILA are set to dashed lines, while AQUILA is still a solid line.
> 2) in Figure 4, all lines are in distinct shapes for better discrimination.
>
> And for the question **''Do they assume the communication TO server only or together with the broadcasting''**, we consider only the communication to server in these figures, and we refer the reviewer to the first question above for the specific concern.

---

### Official Review · Reviewer_9gtf · 2021-11-08

**Correctness:** 3
**Technical Novelty And Significance:** 3
**Empirical Novelty And Significance:** Not applicable
**Recommendation:** 6
**Confidence:** 3

**Main Review:**

Strengths:
1) The paper considers a relevant problem.
2) It is toward unifying the prior work.
3) Changes in the gradients seem a reasonable signal for the adaptation decisions.

Weaknesses:
1) On page 4, the paper mentions that AQUILA has a better method for setting the quantization level as, unlike AdaQuantFL, equation (7) doesn't increase the quantization level over the training process. However, doesn't the size of the gradients (i.e., $||g^k_m||$) decrease during the training process? If so, wouldn't it be better to compare the normalized gradients in eq. (7) rather than the gradients?

2) Have the performance improvements shrunk on the CIFAR10 dataset which is a more complex dataset than MNIST? For example, look at Figure 4 subplots (d) and (f). If so, why does this happen and what is AQUILA's benefit for more complex data distributions in the real world?

**Summary Of The Paper:**

This paper points out that model update frequency and update quantization level are not disjoint decisions in federated learning systems. Furthermore, it explains how to modify and combine prior work on each of the two decisions to develop a new adaptive aggregation method.



**Summary Of The Review:**

I think this paper raises a good question about the interaction between the frequency and quality (i.e., quantization) of model updates in FL. Still, neither the intuitions nor the experiment results convince the reader much about the proposed approach. I'm marginally recommending accepting the paper mainly based on the assumptions that the problem statement is novel and the theoretical methodology is sound to my current understanding of the paper.

---

> ### Author Response · Authors · 2021-11-18
> **Response to Reviewer 9gtf**
>
> Thank you very much for your thoughtful review. We have made revisions to our paper based on your feedback and address your concerns in more detail in the comments below.
>
> **Q1: Doesn't the size of the gradients (i.e., $\|g_{m}^{k}\|$) decrease during the training process? If so, wouldn't it be better to compare the normalized gradients in eq. (7) rather than the gradients?**
>
> **Response:** Thanks for the careful review. First of all, the quantization level given by Eqn.7 is based on the gradient update between iterations instead of the gradient itself, which does not decrease during the training. Besides, the adaptive quantization criterion Eqn.7 has to cooperate with the lazy aggregation strategy Eqn.8, where the communication decision depends on the comparison between gradient changes and the global model updates as well as the quantization error of the original gradients. Given this constraint, the normalization of gradient may induce extra noise to the system, of which the impact needs to be further analyzed in future work.
>
> **Q2: Have the performance improvements shrunk on the CIFAR10 dataset which is a more complex dataset than MNIST? For example, look at Figure 4 subplots (d) and (f). If so, why does this happen and what is AQUILA's benefit for more complex data distributions in the real world?**
>
> **Response:**  Thank you for the constructive comments. For different datasets, it is common for degrees of improvement to vary from one to another. For instance, [R1] evaluates the communication efficiency of its proposed method on MNIST and CIFAR10 dataset like us. In Table 3 of [R1], 73.9\% of communication bits are reduced on MNIST but only 59.3\% are reduced on CIFAR10, demonstrating a similar trend as our experiment results. Besides, we would like to highlight that our proposed AQUILA still outperforms state-of-the-art methods on CIFAR10 dataset even with heterogeneous models and Non-IID data distribution.
>
> *[R1] Ang Li, Jingwei Sun, Binghui Wang, Lin Duan, Sicheng Li, Yiran Chen, and Hai Li. LotteryFL: Personalized and communication-efficient federated learning with lottery ticket hypothesis on non-iid datasets. arXiv:2008.03371, 2020.*
>
> **Q3: I think this paper raises a good question about the interaction between the frequency and quality (i.e., quantization) of model updates in FL. Still, neither the intuitions nor the experiment results convince the reader much about the proposed approach.**
>
> **Response:** Thank you for the professional review. We elaborate the intuition behind our method and the potential impact of our work indicated by our experimental results in a clearer way as follows:
> 1) The upperbound we derive for the expected deviation of lazy aggregation involves a trade-off between the communication frequency (the first term of the right hand side in Eqn.17) and the quantization variance (the second term of the right hand side in Eqn.17). Specifically, the optimization target characterizes the intuition that the decrease of quantization level $b_{m}^{k}$ will lead to more communication rounds given a certain transmission budget, but at the same time more coarse updates for each communication round. Therefore, we build our adaptive quantization criterion based on minimizing this expected deviation for an optimal balance between these two interactive factors during training.
> 2) Fig.3 and Fig.4 verify the effectiveness of our proposed adaptive quantization criterion, where AQUILA achieves good convergence properties with low bits transmission. Meanwhile, Fig.2 and Fig.20 in the revised paper show how the two factors interact with each other. These two figures are consistent with our expectation that the adjustment of the transmission frequency and quantization level can cooperate to further control the overall transmitted burden by skipping high-bit transmission for negligible updates and maintaining a stable relatively-low communication frequency throughout the training.
> 3) Finally, we would like to emphasize that our evaluation results with two datasets of different complexity and a number of non-homogeneous FL scenarios are sufficient to indicate that the proposed AQUILA possesses a convincing superiority in terms of improving communication efficiency as well as compatibility to existing FL settings.
>
> **Q4: Some of the paper’s claims have minor issues. A few statements are not well-supported, or require small changes to be made correct.**
>
> **Response:** Thanks for the careful review. We have carefully checked the claims in the paper and made minor adjustments. Thank you again for your helpful suggestions.

---

### Comment · Area_Chair_CBwp · 2021-12-06
**Extra Review - Part 1**

I am the AC handling this submission. I have read the reviews, author response and scores, which range from mildly supportive (twice a score of 6) to very supportive (one score of 8). I also happen to be expert in the area, and decided to read the paper, which based on the scores seemed very promising, in detail. However, upon completion it was absolutely clear to me that this paper can't be possibly accepted to ICLR as it suffers from multiple issues, some of them smaller, some larger, and some, unfortunately, fatal. Let me elaborate.

1) The paper is written in a very confusing way.

- For example, the authors talk about algorithm (1), but do not say what problem does the algorithm solve. This needs to be inferred from various places in the text. For example, just above equation (1) they mention that $g_m^k = \nabla f_m(D^k_m; \theta^k)$. However, it is not mentioned what the function $f_m$ is, and how do they combine to define the global objective/loss. Also, the dimension of the vector $\theta^k$ is not mentioned at this point. One can only indirectly infer from sentence that includes equation (2), which takes about gradient quantization, and specifies the gradients $g$ as belonging to $\mathbb{R}^d$, that perhaps $\theta^k$ belongs to $\mathbb{R}^d$ also.

- In (4) the authors introduce notation for what they call the quantization error: $\varepsilon_b(g)$. However, this symbol is never defined in the main paper, and one has to search the appendix to find out what the authors mean. Indeed, the first place where it is possible to find an answer to this problem is on page 20, line (15). It is possible to deduce from this place that $\varepsilon_b(g) = Q_b(g)-g$. Admittedly, I guess that this is perhaps what the authors may have meant, but a reader is not supposed to be expected to guess what the authors mean.  A well written paper must be very clear in what notation means what.

- In (6) the authors talk about some function $f$, but this function was never mentioned before. One can only guess that $f$ is perhaps the sum or the average of the functions $f_m$ mentioned before. This guess is only confirmed on page 6 where $f$ is formally introduced. However, this is not the right place for this.

- Equation (4) includes a sum over $D$ elements indexed by $d$. First, this $d$ conflicts with the dimension of $\theta$. Second, $D$ as never introduced. I understand that this is a symbol directly borrow from previous work. However, this was done without mentioning what it means, why it is important, and so on. Same comment can be made about the constants $\xi_1, \dots, \xi_D$ whose nature was never explained beyond saying that they are "some predetermined constant weights". Why are they introduced? What is they purpose? How to choose them. Why?

- The hat versus non-hat notation (e.g., $\hat{\theta}$ vs $\theta$, $\hat{g}$ vs $g$) is also quite confusing. The issue is a bit smaller here, but still, this can and should be clarified substantially.

- Definition 1 is unclear. What does "until a given time instant" mean? This is very unclear and confusing. It is not clear at all what $B_m$ means.

---

### Comment · Area_Chair_CBwp · 2021-12-06
**Extra Review - Part 2**

2) The main theorem (Theorem 1) suffers from the same severe lack of clarity issues. However, it is also quite problematic due to other reasons.

- Clarity: What is $d$? You should introduce the optimization problem in the introduction and clearly state that the parameter is $\theta \in \mathbb{R}^d$, and then this will be OK. However, what is $d_m$? All that is said about this is: "$d_m$ is a positive constant determined by $L_m$". How? Why? What's the point here? A minor point: Say explicitly that $\theta^{k+1}$ is  as defined in (1) and $\hat{\theta}^{k+1}$ is  as defined in (5). What does $\sigma^2$ mean? Saying "$\sigma^2$ denotes the maximum $||w||_2^2$" is far from sufficient. Note that this function us unbounded above. So what do the authors mean?

- Severe issue 1: To find out what  $\sigma^2$ really means, we must go all the way to page 20 in the appendix. Even there this symbol is not defined, just used. But it is used in a way from which we can infer its meaning. In the derivation leading to equation (16) we see that a bound of this form is used: $\mathbb{E} || \varepsilon_b (g) ||^2 \leq \frac{d}{b^2} \sigma^2$. However, using Assumption 2, all that we know is that $\mathbb{E} || \varepsilon_b (g) ||^2 \leq q_n || g ||_2^2$. It is true that for the quantizer the author use, one can bound $q_b \leq \frac{d}{b^2}$. However, there is no a-priori reason to believe that the squared norms of the gradients will be bounded. That is, you can't simply say that $\sigma^2 $ is a constant such that $ || g ||_2^2 \leq \sigma^2$ for all gradients $g$. This is clearly not true if $g$ is allowed to be any vector $g\in \mathbb{R}^d$, and the authors have not presented any proof that the gradients produced by their iterative process will be bounded. This means that Theorem 1, as presented, is meaningless.

- Severe issue 2: The main idea of the paper is to minimize the bound (9) individually for each client as a function of the number of quantization levels $b_m^k$. First, this function can't be treated as a continuous function since the number of levels needs to be a positive integer. So, an argument based on differentiation and setting the result to $0$ is not valid to conclude that a minimizer was found. Second, $(b_m^k)^*$ as written in equation (10) does not even seem to be guaranteed to be larger than 1 (the number of levels needs to be at least 1). Is it? Why? This is not commented on.

- Severe issue 3: Next, why do you minimize (9) in $b_m^k$ and not in other free parameters? For example, the stepsize $\alpha$ is a free parameter that an also be adjusted. Clearly, if one chooses $\alpha=0$, then (9) is minimized. Why is this not a better approach? Clearly, decreasing the stepsize will affect the convergence rate, so one has to be careful. But so will adjusting the number of quantization levels. In fact, this points to a key oversight of this work. In every analysis of GD with quantized or compressed gradients I know of, the optimal stepsize choice *depends* on the properties of the quantizer. For example, if the number of levels increases, the optimal stepisze can also increase. Any one-step analysis of the type performed in this paper will necessarily miss this global conference point of view, and as such will lead to misleading conclusions.

- Sever issue 4: I would like to see a convergence analysis of the proposed method - and more than this, an analysis which shows better convergence rates than the competing methods. No convergence analysis was provided here. The last paragraphs of Section  3 are *not* a convergence analysis. The authors seem to say the analysis follows the exact same steps as prior work. But then no performance gain was proved. Moreover, the authors claim that $L$-smoothness implies that $|| \nabla f(\theta^k) ||_2^2 \leq 2L [f(\theta^k) - f(\theta^*)]$. This is *not true*. Convexity is needed as well. Convexity is never assumed anywhere in the paper.

- Sever issue 5: Theorem 2 introduces constants $\xi$ and $P$ which are not defined nor commented on. How should they be set? Would not the stepsize bound be too severe then? If Theorem 2 only holds for a very small stepsize, and that seems to be the case, it is meaningless because such stepsizes will not be used in practice. How does this stepsize compare to that used by GD with quantized gradients (and no adaptivity or lazy aggregation?). Do we gain or lose? How much? To my mind, Theorem 2 is meaningless. The claim that "the proposed AQUILA is proven to be more efficient ..." is meaningless at best, and false at worst. What needs to be performed is a global convergence analysis - and what needs to be argued is that the proposed methods will converge to the solution in fewer communicate bits. This was not done. The entire paper focuses on a one-step analysis.

---

### Comment · Area_Chair_CBwp · 2021-12-06
**Extra Review - Part 3**

3. The authors do not compare against the state of the art methods that use communication compression. In the strongly convex case, accelerated methods exist. Granted, they do not use adaptive quantization, nor lazy aggregation. But the combination of acceleration and quantization with a fixed number of quantization levels is very powerful, and currently the state of the art in theory.

4. In the nonconvex case, the current theoretical SOTA methods rely on a much more sophisticated method: on compressing gradient differences (see the MARINA method of Gorbunov et al; ICML 2021). AQUILA does not do that, and hence can't compete in theory. The difference between MARINA and QSGD (which is the baseline method the authors are essentially trying to improve on via adaptive quantization and lazy aggregation) for nonconvex problems is massive. I do not believe that AQUILA can be better in practice. However, my belief does not matter: a comparison is needed.

5. Th authors use a simple version of the quantization operator from 2017. A better quantizer was proposed by Horvath et al (arXiv:1905.10988, 2019) - one whose $q_b$ parameter is *exponentially* smaller given the same number of levels $b$. So, the authors do not build on the latest results in the area of quantization either.

6. I noticed many issues with the experiments. But I mentioned enough before, so I will not comment on this part of the paper much.


---

In short, this paper has many issues, some of them severe, and a few fatal. As such, I cannot recommend it for acceptance to ICLR.

The Area Chair

---

### Decision · Program_Chairs · 2022-01-20

**Decision:**

Reject

**Comment:**

Two of the initial reviews of the paper were mildly positive (2 scores of 6), and one was very positive (score of 8). However, these reviews failed to notice some severe issues with the paper, which were detailed by the Area Chair in an Extra Review which was provided late. The severe issues include: clarity of exposition (undefined notation in many places) and theory (vacuous or meaningless theorems and assumptions). I apologize to the authors for not having had the chance to defend against this late review. However, the issues are indeed severe.